# Perilipin 5 links mitochondrial uncoupled respiration in brown fat to healthy white fat remodeling and systemic glucose tolerance

Violeta I. Gallardo-Montejano[1], Chaofeng Yang[1], Lisa Hahner[1], John L. McAfee [1,4], Joshua A. Johnson[2], William L. Holland[2,5], Rodrigo Fernandez-Valdivia[3] & Perry E. Bickel [1✉]

Exposure of mice or humans to cold promotes significant changes in brown adipose tissue (BAT) with respect to histology, lipid content, gene expression, and mitochondrial mass and function. Herein we report that the lipid droplet coat protein Perilipin 5 (PLIN5) increases markedly in BAT during exposure of mice to cold. To understand the functional significance of cold-induced PLIN5, we created and characterized gain- and loss-of-function mouse models. Enforcing PLIN5 expression in mouse BAT mimics the effects of cold with respect to mitochondrial cristae packing and uncoupled substrate-driven respiration. PLIN5 is necessary for the maintenance of mitochondrial cristae structure and respiratory function during cold stress. We further show that promoting PLIN5 function in BAT is associated with healthy remodeling of subcutaneous white adipose tissue and improvements in systemic glucose tolerance and diet-induced hepatic steatosis. These observations will inform future strategies that seek to exploit thermogenic adipose tissue as a therapeutic target for type 2 diabetes, obesity, and nonalcoholic fatty liver disease.

[1] Division of Endocrinology, Department of Internal Medicine, The University of Texas Southwestern Medical Center, Dallas, TX, USA. [2] Touchstone Diabetes Center, Department of Internal Medicine, The University of Texas Southwestern Medical Center, Dallas, TX, USA. [3] Department of Pathology, Wayne State University School of Medicine Detroit, Detroit, MI, USA. [4] Present address: Pathology and Laboratory Medicine Institute, Cleveland, OH, USA. [5] Present address: Department of Nutrition and Integrative Physiology and the Diabetes and Metabolism Research Center, University of Utah, Salt Lake City, UT, USA. ✉email: perry.bickel@utsouthwestern.edu

Brown adipose tissue (BAT) is a thermogenic organ in mammals that generates heat to maintain body temperature during exposure to cold ambient temperature[1]. In addition to brown adipocytes, "brown-like cells" called beige[2,3] or brite[4] thermogenic adipocytes are recruited in white adipose tissue (WAT) depots in response to diverse stimuli, including chronic β-agonist treatment, cold exposure, and exercise. In many thermogenic adipocytes, expression of uncoupling protein 1 (UCP1) and its activation by fatty acids drive heat production by promoting the leak of protons across the inner mitochondrial membrane, thereby uncoupling oxidative phosphorylation from the production of ATP[5,6].

Over the past 10 years, the presence of thermogenic adipose tissue in adult humans has been reported[7–10]. The potential of cold-activated, UCP1-positive adipocytes in humans as a target for the treatment of metabolic diseases, such as diabetes and obesity, is the subject of recent reviews[11–15], and has emerged as an important field of study[16–19]. In both humans and mice, BAT activation, for example, by exposure to cold, leads to increased uptake of both glucose and fatty acids in BAT, as well as to improved systemic glucose tolerance and insulin sensitivity[7–10,20–24]. Even though important advances have elucidated many aspects of how BAT functions, many questions remain unanswered, including how mitochondrial capacity within thermogenic adipocytes is regulated to accommodate environmental, nutritional and hormonal signals, or stresses.

Perilipin 5 (PLIN5) is a member of the Perilipin family of lipid droplet (LD) proteins that is expressed in highly oxidative tissues, such as heart, oxidative skeletal muscle, fasted liver, and adipose tissues[25–27]. PLIN5 gain-of-function in COS-7 cells and OP9 preadipocytes promotes increased triacylglycerol (TAG) storage and fatty acid oxidation[25]. Additional studies in cultured cells overexpressing PLIN5 have suggested a role for PLIN5 in the regulation of lipolysis via protein–protein interactions with adipose triglyceride lipase (ATGL) and the activator of ATGL, 1-acylglycerol-3-phosphate O-acyltransferase (ABHD5)[28,29]. PLIN5 has been localized to muscle mitochondria[30], and may play a role in promoting physical interactions between mitochondria and LDs[31]. In adipose tissues, PLIN5 is expressed in inguinal and epididymal WAT (iWAT and eWAT, respectively), but its highest expression among adipose depots occurs in BAT[25]. Data from genetically engineered mouse models have suggested that PLIN5 influences systemic metabolism. Constitutive PLIN5 total body knockout mice are insulin resistant due to reduced glucose disposal in muscle and WAT[32]. Consistent with these data, in a group of 85 nondiabetic human subjects, PLIN5 RNA expression in subcutaneous adipose tissue negatively correlated with body mass index and positively correlated with insulin sensitivity[25]. In another study of human subjects, PLIN5 expression in abdominal subcutaneous adipose tissue of Finnish males positively correlated with favorable metabolic traits, and negatively with adverse metabolic traits[33]. Collectively, these data from mice and humans suggest an important role for PLIN5 in the metabolic functions of adipose tissues. Previously, we demonstrated in cell culture models that during catecholamine-stimulated lipolysis PLIN5 forms transcriptional complexes with PGC1α in the nucleus, and promotes transcription of genes involved in mitochondrial biogenesis and oxidative metabolism, including the gene encoding PGC1α[34]. In the same study, we demonstrated that PLIN5 silencing in a BAT cell line is associated with decreased mitochondrial respiration and reduced thermogenic gene expression[34]. However, the functions of PLIN5 specifically in BAT or WAT have not been studied in vivo.

We hypothesized that PLIN5 functions in BAT to augment mitochondrial respiratory and thermogenic capacity in response to increased metabolic demand, for example, during cold exposure in mice. To test this hypothesis, we generated mouse models of gain and loss of PLIN5 function, specifically in BAT. Here, we report that PLIN5 protein increases in BAT dramatically within 48 h of cold exposure and that PLIN5 augments fatty acid uptake, mitochondrial biogenesis, cristae packing, and oxidative function in BAT. In addition, we show that the effects of PLIN5 in BAT secondarily lead to smaller adipocytes in iWAT, to improved systemic glucose tolerance and insulin sensitivity, and to protection from hepatic steatosis on a high-fat diet (HFD).

## Results

**Perilipin 5 protein and gene expression are induced by cold exposure and β₃ adrenergic agonist in brown adipose tissue.** To test whether PLIN5 expression is regulated during the activation of BAT, three groups of 12-week-old male mice (C56BL/6 J) housed at 23 °C were shifted to 6 °C or to 30 °C or were left at 23 °C (Fig. 1a). After 16 h, we assessed interscapular BAT for PLIN5 RNA and protein. RNA for PLIN5 in BAT was increased threefold in mice exposed to 6 °C compared to mice shifted to 30 °C and increased twofold compared to mice maintained at 23 °C (Fig. 1b). Relative PLIN5 protein in the BAT of these three test groups was consistent with the relative RNA expression (Fig. 1c), and the level of PLIN5 protein rose in parallel to that of UCP1 as temperature decreased. To assess the induction of PLIN5 protein in BAT over a longer period of cold exposure, we assessed PLIN5 protein levels by immunoblotting at intervals from 2 h to 7 days (Fig. 1d). PLIN5 protein increased significantly by 12 h at 6 °C and reached a maximum at 48 h (Fig. 1e). In fact, PLIN5 protein in BAT at 48 h of cold exposure was 40-fold greater than PLIN5 protein after 7 days at thermoneutrality (30 °C; Fig. 1e, right panel).

In addition to exposure to cold, the pharmacological selective activation of β₃ adrenergic receptors activates thermogenesis in BAT[1]. We tested if BAT PLIN5 protein and Plin5 mRNA expression increases with the administration of a selective β₃ adrenergic agonist (CL-316,243). We intraperitoneally injected 12-week-old male mice (C56BL/6 J) with either Vehicle or CL-316,243 daily for 2 days or 7 days. We observed an increase in PLIN5 protein levels (Supplementary Fig. 1a) and Plin5 mRNA (Supplementary Fig. 1b) with CL-316,243 at both time points with the highest levels at 2 days of treatment, which was similar to our findings after shifting mice to housing at 6 °C.

**Generation of doxycycline-inducible, brown adipocyte-specific Perilipin 5 overexpression mice (BATiPLIN5).** The dramatic responsiveness of PLIN5 protein in BAT to ambient temperature or pharmacological activation of β3 adrenergic receptors suggested that its level of expression may influence critical BAT functions during cold stress. To investigate PLIN5 gain-of-function in BAT in vivo, we used the tetracycline-controlled transcriptional activation system. Inducible PLIN5 expression was conferred by a novel TRE-Plin5 allele in the C56BL/6J strain background (Fig. 1f, see "Methods" for details). The mice were bred to produce two test groups: mice carrying both the TRE-Plin5 allele and the UCP1rtTA allele, termed the "BATiPLIN5" strain, and Control mice carrying only the UCP1rtTA allele. To test PLIN5 inducibility and tissue specificity, both BATiPLIN5 mice and Control mice housed at 23 °C were fed normal chow containing doxycycline (DOX) (200 mg DOX/kg chow) for 4 weeks, and then tissues were harvested for measurement of Plin5 mRNA by quantitative real-time PCR (qPCR). BATiPLIN5 mice expressed tenfold greater Plin5 RNA in BAT compared with Control mice, but there were no differences in Plin5 RNA between these strains in other tissues (Fig. 1g). In addition, in mice housed at 23 °C we observed a 4.5-fold increase in Perilipin 5 protein levels in the BAT of BATiPLIN5 compared with

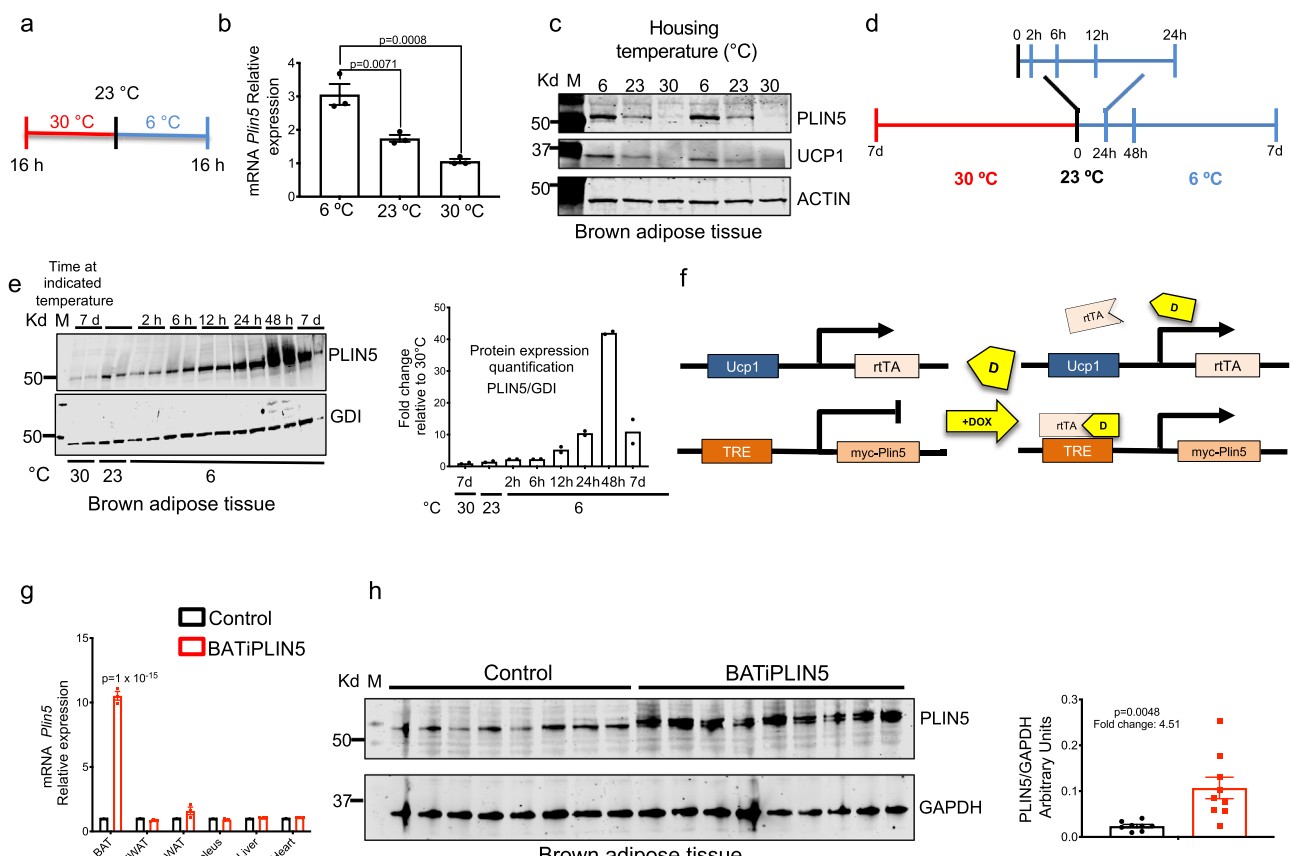

**Fig. 1 PLIN5 expression in brown adipose tissue is regulated by temperature and generation and validation of inducible, BAT-specific PLIN5 transgenic mice (BATiPLIN5). a** Timeline for controlled temperature experiments depicted in **b** and **c**. We collected samples for RNA and western blot (WB) at the indicated time points. **b** qPCR for *Plin5* relative mRNA expression in brown adipose tissue (BAT) from wild-type mice (C56BL/6J) housed at the indicated temperatures. $n = 3$ mice per group. **c** WB for PLIN5 in BAT from wild-type mice (C56BL/6 J) housed at the indicated temperatures. **d** Timeline for controlled temperature experiment depicted in **e**. **e** BAT WB for PLIN5 from wild-type C56BL/6 J mice housed at the indicated temperatures (left panel) and WB quantification of PLIN5 normalized to GDI (right panel). $n = 2$ mice per group. **f** Schematic representation for doxycycline-inducible myc-PLIN5 overexpression mice in BAT (BATiPLIN5 mice). **g** qPCR for *Plin5* relative mRNA expression from the indicated tissues from Control or BATiPLIN5 mice housed at 23 °C. $n = 3$ mice per group. **h** WB for PLIN5 from BAT of Control ($n = 8$ mice) or BATiPLIN5 ($n = 9$ mice) housed at 23 °C (left panel). WB quantification of PLIN5 normalized to GAPDH (right panel). Values are mean ± s.e.m. For **b**, statistical analysis was performed using one-way ANOVA followed by Tukey posttest. For **g** and **h**, statistical analysis was performed using unpaired two-sided Student's *t* test. Source data are provided as a Source data file.

Control mice (Fig. 1h). Is important to note that due to the low activity at 30 °C of the Ucp1 promoter used to express rtTA, we did not observe increased expression of PLIN5 in the BAT of mice housed at thermoneutrality (Supplementary Fig. 2).

**Perilipin 5 overexpression in BAT improves glucose tolerance and insulin sensitivity at room temperature with reduction in weight only during cold exposure.** Given our data that PLIN5 promotes mitochondrial function in a brown adipocyte cell line[34], we asked whether PLIN5 gain-of-function in BATi-PLIN5 mice would affect body weight, body composition, and/or glucose homeostasis. Both Control and BATiPLIN5 mice showed similar body weight after 25 weeks on chow diet and both gained weight on HFD with no differences between genotypes (Fig. 2a). All mice were fed diet that contained 200 mg DOX/kg diet starting at 8 weeks of age. After 8 weeks on the indicated diets, body composition and food intake were similar in both groups of mice housed 23 °C (Fig. 2b, c). Energy expenditure and multidimensional activity in mice fed chow diet and housed at 23 °C were similar in both groups (Supplementary Fig. 3). In contrast, when housed at 6 °C and fed chow diet, BATiPLIN5 mice had greater food intake compared

with Control mice (Fig. 2d), but nevertheless lost more weight (Fig. 2e). Random blood glucose levels in mice housed at 23 or 30 °C overnight were not significantly different between groups, but BATiPLIN5 mice housed overnight at 6 °C showed lower random blood glucose levels than Control mice (Fig. 2f). We next performed oral glucose tolerance tests (OGTTs) on mice fed HFD for 8 weeks and housed at 23 °C. BATiPLIN5 mice had improved glucose tolerance with lower insulin levels compared with Control mice (Fig. 2g). BATiPLIN5 mice also showed lower levels of glucose during insulin tolerance testing (Fig. 2h). We found similar results for OGTT and insulin tolerance test in mice fed chow diet (Fig. 2i, j).

**Perilipin 5 overexpression in BAT increases cold tolerance and thermogenic gene expression in brown adipose tissue.** To assess whether increased expression of PLIN5 would improve the ability to maintain body temperature during cold challenge, we used implantable temperature probes to measure body temperature of BATiPLIN5 and Control mice during cold exposure at 6 °C, both acutely without access to food and chronically with food. The timeline for these experiments is outlined in Fig. 3a. For acute cold assessment, we transferred mice from 23 to 6 °C and withdrew food.

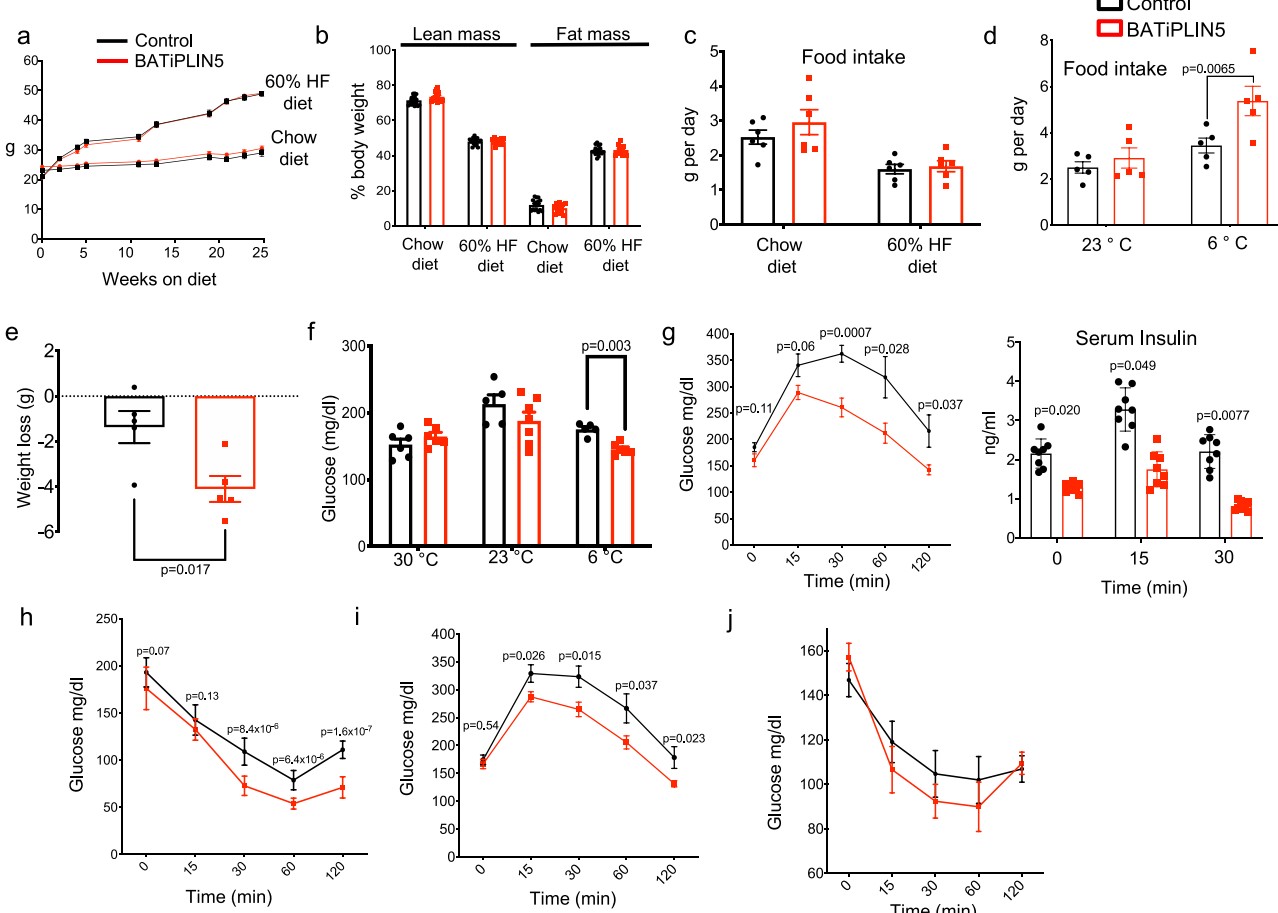

**Fig. 2 PLIN5 overexpression in BAT improves glucose tolerance and insulin sensitivity at room temperature with changes in weight and blood glucose only during cold exposure (data for Control mice are in black and for BATiPLIN5 mice in red). a** Body Weight at the indicated time points of Control or BATiPLIN5 mice that were fed chow (Control $n = 27$, BATiPLIN5 $n = 23$ mice) or HF diet (Control $n = 12$, BATiPLIN5 $n = 16$ mice). **b** Mass composition after 8 weeks on Chow or HF diet. (Control $n = 15$, BATiPLIN5 $n = 16$ mice for each diet). **c** Daily food intake measurement during 6-day period in mice housed at 23 °C. Before measurement mice were fed for 8 weeks with chow or HF diet. $n = 6$ mice per group. **d** Daily food intake during 6-day period in mice housed at 23 or 6 °C. Before exposure to cold mice were fed for 8 weeks with chow diet and housed at 23 °C. $n = 5$ mice per group. **e** Weight loss in mice housed at 6 °C for 15 days. $n = 5$ per group. **f** Blood glucose of mice housed at 23 °C or exposed to 30 or 6 °C for 24 h. Glucose was measured in fed state at 9 a.m. $n = 8$ mice per group. **g** Oral glucose tolerance test (OGTT) after 8 weeks on HF diet. Glucose (left panel) and insulin levels (right panel) at the indicated time points. $n = 8$ per group. **h** Insulin tolerance test (ITT) after 8 weeks on HF diet ($n = 10$ Control $n = 9$ BATiPLIN5 mice). **i** OGTT after 8 weeks on chow diet. Glucose levels at the indicated time points ($n = 15$ Control $n = 16$ BATiPLIN5 mice). **j** Insulin tolerance test (ITT) after 8 weeks on chow diet. Glucose levels at the indicated time points. ($n = 15$ Control $n = 16$ BATiPLIN5 mice). Data are presented as means ± s.e.m. For **a–i**, we used unpaired two-sided Student's *t* test for statistical analysis. Source data are provided as a Source data file.

We measured body temperature every hour for 8 h. Both Control and BATiPLIN5 mice showed a decrease in body temperature with no differences between genotypes during the first 6 h. However, at 8 h the body temperature of Control mice continued to fall, while that of BATiPLIN5 mice began to increase (Fig. 3b). At 8 h, the acute cold exposure experiment ended, and ad libitum access to food was restarted. Both study groups were maintained at 6 °C, and body temperature was measured in the fed state at 9 a.m. on days 2, 4, 14, and 21. Body temperature remained higher in BATiPLIN5 mice than Control mice at each timepoint (Fig. 3c). For cold acclimation, on the morning of day 22, we withdrew food and measured body temperature every hour for 7 h. By 5 h without food, BATiPLIN5 mice began to maintain higher body temperatures than Control mice (Fig. 3d). We next investigated thermogenic gene expression in the BAT of mice housed at 23 °C or exposed for 16 h to 6 °C. We found that expression of some thermogenic genes was increased in BATiPLIN5 mice compared with Control mice at both 23 and 6 °C (Fig. 3e).

**Perilipin 5 overexpression in BAT increases fatty acid uptake and oxidation in brown adipose tissue.** Cardiac-specific overexpression of PLIN5 is associated with myocardial steatosis[35,36], thought to be due to its function on the LD as a lipolytic barrier[36]. Similarly, PLIN5 overexpression in the liver leads to hepatic steatosis[37], and PLIN5 liver knockout reduces liver triglycerides, as well as the size and number of LD[38]. We investigated whether PLIN5 overexpression in BAT would result in increased lipid accumulation in this tissue. First, we performed conventional hematoxylin and eosin (H&E) histology of BAT harvested from BATiPLIN5 and Control mice exposed to 30, 23, and 6 °C for 16 h. We found no differences in histology at 30 °C between genotypes (Supplementary Fig. 4). However, in BATiPLIN5 mice housed at 23 °C, we observed a shift to smaller LDs when compared with Control mice (Fig. 4a), and this shift was more accentuated in mice housed at 6 °C (Supplementary Fig. 4). We found no differences in BAT weight (Fig. 4b, left panel) or BAT TAG content between BATiPLIN5 and Control mice housed at 23 °C (Fig. 4b, right panel).

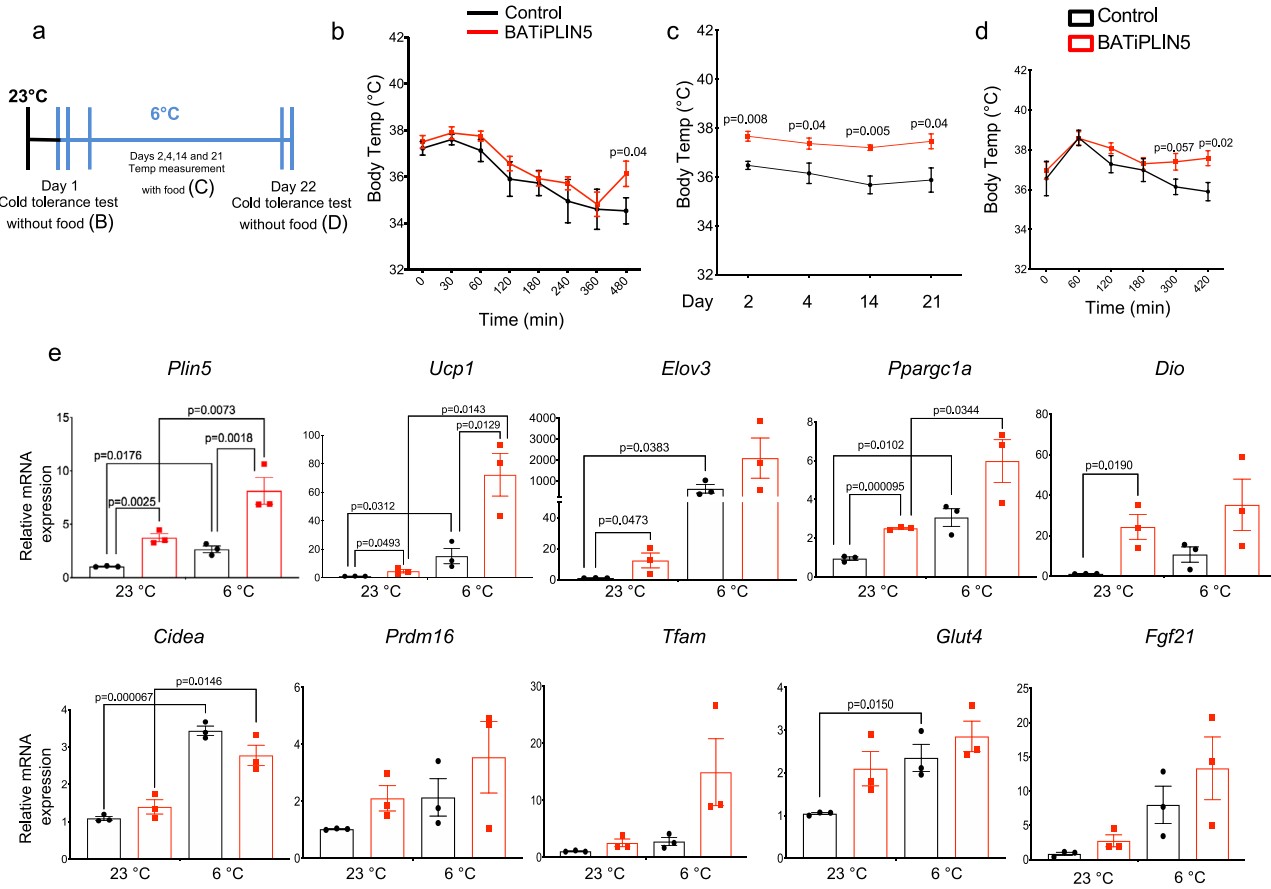

**Fig. 3 PLIN5 overexpression in BAT increases cold tolerance, cold acclimation, and thermogenic gene expression in BAT. a** Timeline for cold tolerance experiments for **b**–**d**. Mice were housed at 23 °C before cold tolerance experiments. **b** Body temperature at the indicated time points during acute cold tolerance test. $n = 10$ mice per group. **c** Body temperature in fed mice housed at 6 °C on the indicated days. $n = 10$ mice per group. **d** Body temperature at the indicated time points during acclimation cold tolerance test. $n = 10$ mice per group. **e** Relative mRNA expression by qPCR for the indicated genes in BAT from Control or BATiPLIN5 mice housed at 23 °C or exposed 6 °C for 16 h. $n = 3$ mice per group. Data are presented as means ± s.e.m. For **b**–**d**, unpaired two tailed Student's $t$ test was used for statistical analysis. For **e**, statistical analysis was performed using two-ANOVA followed by Tukey posttest. $P$ values are shown in the figure. Source data are provided as a Source data file.

However, the TAG content of the BAT from BATiPLIN5 mice housed at 6 °C overnight (Supplementary Fig. 4 and Fig. 4b, right panel) was greater than that from Control mice without any differences in BAT weight between the genotypes (Fig. 4b, left panel). Due to the above and previous reports that PLIN5 regulates ATGL and lipolysis[28,29], and can act as a barrier to lipases[38], we next investigated whether overexpression of PLIN5 in BAT affects BAT lipolysis. We measured ex vivo basal and β3 adrenergic-stimulated lipolysis in BAT explants, and found no differences between the genotypes (Fig. 4c).

To assess whether BAT overexpression of PLIN5 affects circulating lipids, we measured fasting serum triglycerides and nonesterified free fatty acids (NEFA), and found a trend of lower NEFA in the BATiPLIN5 mice compared with Control mice on chow, but not HFD (Fig. 4d) and a trend to lower NEFA in fed mice exposed overnight to cold (Fig. 4e). We next challenged BATiPLIN5 or Control mice with a triglyceride load by oral gavage of olive oil and measured serum triglycerides at several time points. BATiPLIN5 mice showed improved triglyceride clearance when compared with Control mice (Fig. 4f). We next tested lipid clearance and tissue uptake and oxidation by injection of radiolabeled triglycerides, and found that triglyceride clearance (Fig. 4g) was increased in the BATiPLIN5 mice. Also, fatty acid uptake (Fig. 4h) and oxidation (Fig. 4i) were increased in the BAT of BATiPLIN5 mice relative to Control mice. To investigate

possible mechanisms of increased fatty acid uptake in BAT, we measured gene expression of *Cd36*, *Lpl*, *Angplt4*, and *Fatp1*. Of these genes, only expression of *Lpl* was increased significantly in BATiPLIN5 mice relative to Controls, and only after overnight cold exposure (Fig. 4j). LPL is rate-limiting for triglyceride clearance from plasma and for tissue fatty acid uptake[39], including in BAT[24]. We measured glucose uptake after oral gavage of radiolabeled $^{14}$C-deoxyglucose in mice housed at 23 °C. We found no differences between glucose uptake into the BAT of Control and BATiPLIN5 mice or in any of the other tissues studied (Supplementary Fig. 5a).

In addition to circulating lipids and glucose, de novo lipogenesis in BAT is an important mechanism to fuel BAT thermogenesis[40–42]. We therefore measured expression of genes involved in de novo lipogenesis in BATiPLIN5 and Control mice housed at 23 °C and after overnight exposure to 6 °C. We found increased expression of the lipogenic genes *Acss2* (acetyl-CoA synthetase) and *Acaca* (acetyl-CoA carboxylase 1) in BATiPLIN5 mice compared with Control mice, as well as trends to increased expression of *Fasn* (fatty acid synthase) and carbohydrate response element-binding protein beta (*Mlxipl*, aka *Chrebp*). These differences in gene expression between genotypes were observed at normal housing temperatures, but not after acute cold exposure (Supplementary Fig. 5b).

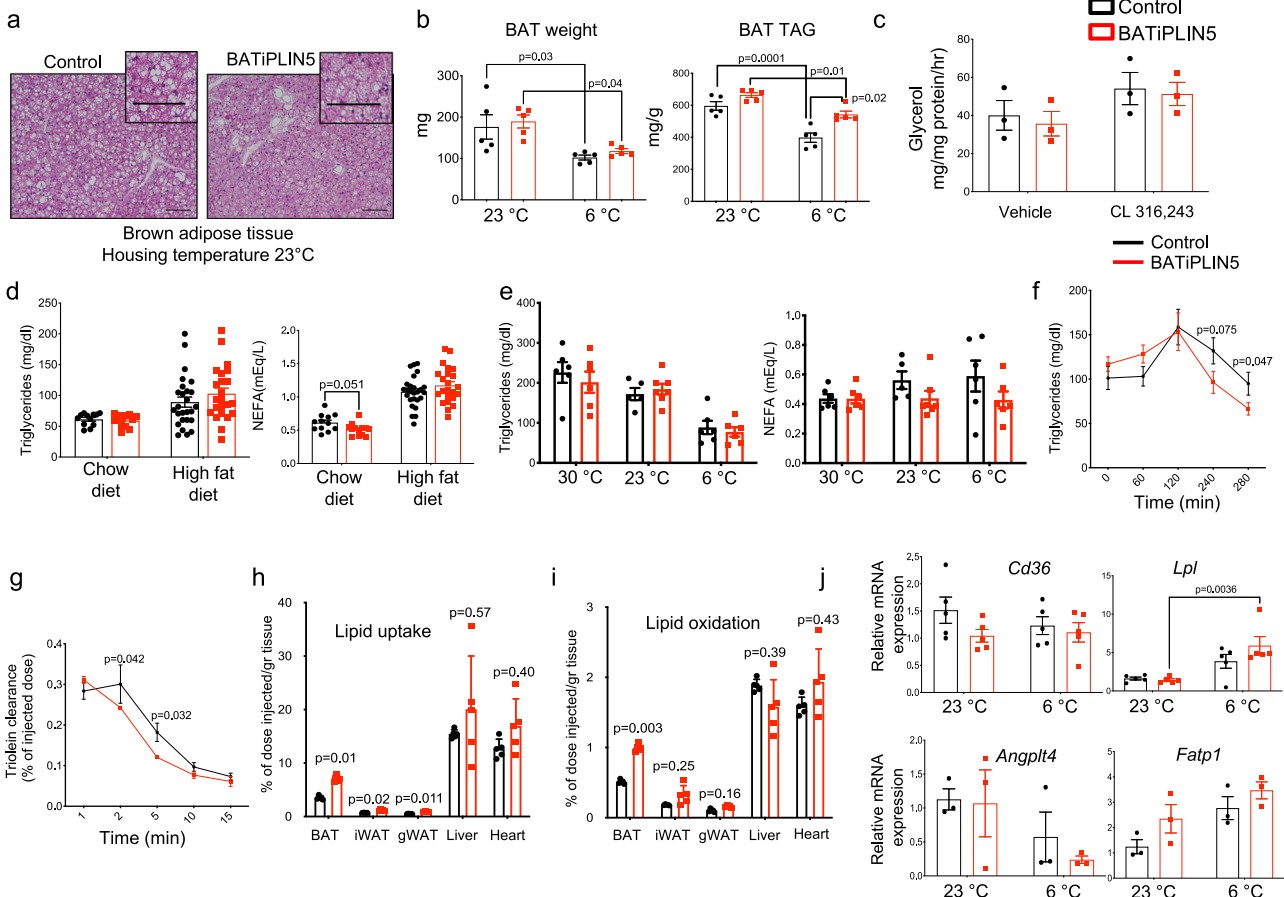

**Fig. 4 PLIN5 overexpression in BAT promotes fatty acid uptake and oxidation in BAT. a** Representative images of hematoxylin and eosin staining of BAT from Control or BATiPLIN5 mice housed at 23 °C. Scale bar = 100 μm. **b** BAT weight and triglyceride content from Control or BATiPLIN5 mice housed at 23 °C or exposed to 6 °C overnight. $n = 5$ mice per group. **c** Ex vivo lipolysis assay (glycerol release) performed on minced BAT harvested from Control or BATiPLIN5 mice, and treated with Vehicle or β3 adrenergic agonist CL-316,243. $n = 3$ mice per group. **d** Serum triglycerides (left panel) and serum nonesterified fatty acids (NEFA; right panel) from overnight fasted Control or BATiPLIN5 mice fed Chow or 60% HF diet. Chow diet $n = 12$ mice per group. HF diet: Control $n = 26$ mice BATiPLIN5 $n = 23$ mice. **e** Serum triglycerides (left panel) and NEFA (right panel) of Control or BATiPLIN5 mice housed at 23 °C or exposed to 30 or 6 °C for 24 h. $n = 8$ per group. **f** Serum triglyceride clearance after olive oil oral gavage from Control ($n = 12$ mice) or BATiPLIN5 mice ($n = 16$ mice). **g** Serum triolein [9,10-3H(N)]-clearance after iv injection from Control or BATiPLIN5 mice. $n = 3$ mice per group. **h** Triolein [9,10-3H (N)]-uptake in indicated tissues from Control or BATiPLIN5 mice. $n = 5$ mice per group. **i** Triolein [9,10-3H(N)]-oxidation in indicated tissues from Control or BATiPLIN5 mice. $n = 5$ mice per group. **j** Relative mRNA expression by qPCR for the indicated genes in BAT from Control or BATiPLIN5 mice housed at 23 °C or exposed 6 °C for 16 h. *Cd36* and *Lpl* $n = 5$ mice per group. *Angplt4* and *Fatp*1 $n = 3$ mice per group. Values are mean ± s.e.m. For **b** and **j**, statistical analysis was performed using two-way ANOVA followed by Tukey posttest. For **d**, **f**–**i**, statistical analysis was performed using unpaired two-sided Student's *t* test. *P* values are shown in the figure. Source data are provided as a Source data file.

**Perilipin 5 overexpression in BAT is associated with increased insulin sensitivity and decreased inflammation in iWAT.** As reported above, BATiPLIN5 mice are more glucose tolerant and insulin sensitive, but we are unable to explain the improved glycemia by the mechanism of increased glucose uptake by BAT. Liver and WAT are systemic regulators of glucose metabolism and insulin action[43–48]. For this reason, we next investigated whether BAT PLIN5 overexpression exerts secondary effects on iWAT or liver. Hypertrophic white adipocytes have been associated with unhealthy expansion of iWAT that may result in glucose intolerance and insulin resistance[49]. To assess adipocyte size in BATiPLIN5 and Control mice, we performed H&E staining of iWAT. We observed a decrease in adipocyte size in the BATiPLIN5 relative to Control mice housed at 23 °C (Fig. 5a left panel, quantification right panel). These differences in adipocyte size were more marked in mice exposed to cold (Supplementary Fig. 6a). We did not observe an increase in multilocular LDs that would suggest beiging of iWAT; on the contrary, iWAT

expression of beiging genes was decreased in the BATIiPLIN5 mice (Fig. 5b). To assess the degree of beiging by the extent of sympathetic innervation of iWAT, we measured gene expression of tyrosine 3-hydroxylase (*Th*) and dopamine beta-hydroxylase (*Dbh*), which are enzymes required for synthesis of nor-epinephrine. As expected, we found an increase of mRNA for these enzymes in both Control and BATiPLIN5 mice exposed to 6 °C compared to 23 °C, but no differences between these genotypes at either temperature (Supplementary Fig. 7).

Wolins' group reported that transgenic mice with PLIN5 overexpression in skeletal muscle display increased glucose tolerance and resistance to hepatic steatosis, which they attributed to increased serum levels of FGF21 (ref. [50]). We therefore measured serum levels of FGF21 in the Control and BATiPLIN5 mice, but there were no differences in serum FGF21 between the genotypes (Supplementary Fig. 8a).

Another characteristic of unhealthy WAT expansion is a pro-inflammatory state denoted by increased chemotactic signals for

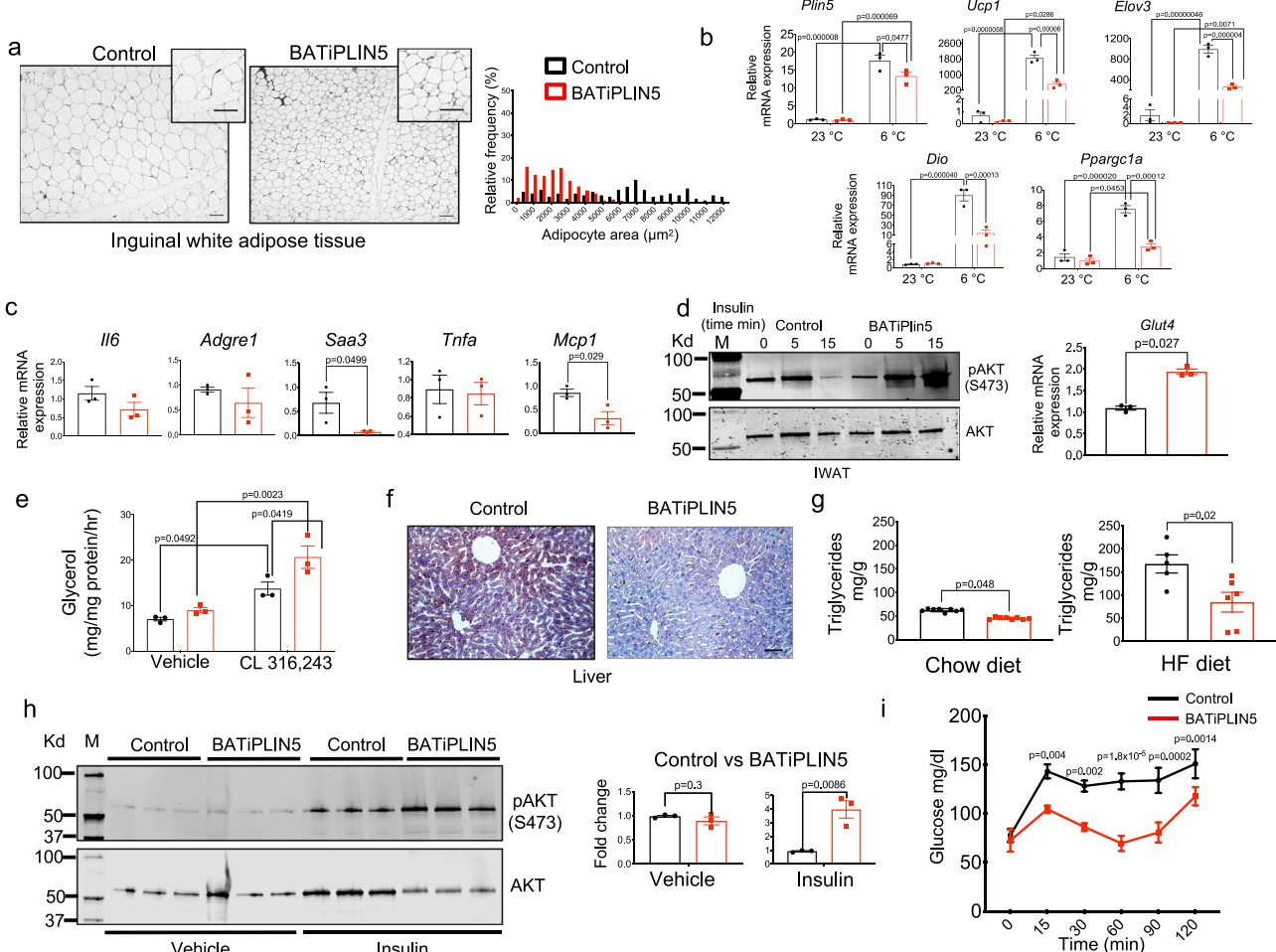

**Fig. 5 PLIN5 overexpression in BAT promotes healthy remodeling of iWAT and protects against diet-induced hepatic steatosis. a** Representative image of hematoxylin and eosin staining of iWAT from Control or BATiPLIN5 mice housed at 23 °C (left panel) and adipocyte area percentage of frequency distribution (right panel). Scale bar = 100 μm. **b** Relative mRNA expression by qPCR for the indicated thermogenic genes in iWAT from Control or BATiPLIN5 mice housed at 23 °C or exposed 6 °C for 16 h. $n = 3$ mice per group. **c** Relative mRNA expression qPCR for the indicated genes in iWAT from Control or BATiPLIN5 mice housed at 23 °C. $n = 3$ mice per group. **d** Left panel: WB from iWAT harvested from Control or BATiPLIN5 mice blotted for phosphorylated AKT (S473) and total AKT. Right panel: relative mRNA expression of *Glut4* ($n = 3$ mice per group) by qPCR of iWAT from Control or BATiPLIN5 mice housed at 23 °C. **e** Ex vivo lipolysis assay (glycerol release) performed on minced iWAT harvested from Control or BATiPLIN5 mice and treated with Vehicle or β3 adrenergic agonist CL-316,243. $n = 3$ mice per group. **f** Representative image of liver Oil red O staining from fasted Control mice or BATiPLIN5 mice fed HF diet for 12 weeks. Scale bar = 100 μm. **g** Liver TAG content from BATiPLIN5 or Control mice fed with chow diet (left panel, $n = 8$ per group) or HF diet (right panel, Control $n = 5$ and BATiPLIN5 $n = 6$) for 8 weeks. **h** WB of liver from Control or BATiPLIN5 mice depicting phosphorylated AKT (S473) and total AKT. Mice were intraperitoneally injected with insulin, and liver was harvested 15 min after insulin injection (left panel) WB quantification (pAKT/AKT ratio fold change; right panel). **i** Pyruvate tolerance test performed on Control or BATiPLIN5 mice housed at 23 °C. $n = 4$ mice per group. Values are mean ± s.e.m. For **c, d, g–i**, statistical analysis was performed using unpaired two-sided Student's $t$ test. For **b** and **e**, statistical analysis was performed using two-way ANOVA followed by Tukey posttest. $P$ values are shown in the figure. Source data are provided as a Source data file.

recruitment of macrophages into WAT[51–53]. We observed a significant decrease in iWAT gene expression of serum amyloid A3 protein (*Saa3*) and monocyte chemoattractant protein 1 (*Mcp1*) in the BATiPLIN5 mice (Fig. 5c). We next investigated insulin signaling by assessing basal and insulin-stimulated AKT phosphorylation in the iWAT of BATiPLIN5 mice and Control mice. We prepared lysates of iWAT freshly harvested from fasted mice at time 0 and at 5 and 15 min after intraperitoneal insulin injection, and found increased insulin-stimulated pAKT (S473) in BATiPLIN5 mice relative to Control mice (Fig. 5d, left panel), as well an increase in *Glut4* mRNA expression in the BATiPLIN5 mice (Fig. 5d, right panel). We measured ex vivo basal and β₃ adrenergic-stimulated lipolysis in iWAT explants, and found that β-3 agonist-stimulated lipolysis in iWAT was increased in

BATiPLIN5 mice (Fig. 5e). Because *Plin5* mRNA expression does not differ in gonadal WAT between BATiPLIN5 and Control mice housed at either 23 or 6 °C, we did not further evaluate the phenotype of gWAT (Supplementary Fig. 8b).

Diet-induced obesity and insulin resistance are risk factors for nonalcoholic fatty liver disease[54]. Consistent with the improvement in systemic insulin resistance and in iWAT insulin sensitivity that we observed in BATiPLIN5 mice, the livers of these mice showed significantly less steatosis than Control mice after 8 weeks of HFD, as assessed by oil red O staining (Fig. 5f), H&E staining (Supplementary Fig. 9), and biochemical TAG extraction and quantification (Fig. 5g). Protection of BATiPLIN5 mice from HFD-induced hepatic steatosis was associated with increased hepatic insulin-stimulated AKT phosphorylation at

serine 473 (Fig. 5h, WB left panel, quantification right panel). Insulin suppresses hepatic gluconeogenesis through several mechanisms[44]. We performed pyruvate tolerance tests as an indirect method to assess whether increased hepatic insulin sensitivity was resulting in the reduced hepatic gluconeogenesis. BATiPLIN5 mice had lower levels of blood glucose after the pyruvate injection than Control mice, which suggests that hepatic gluconeogenesis is reduced in BATiPLIN5 mice (Fig. 5i).

**Perilipin 5 regulates BAT mitochondrial form and function in a UCP1-dependent manner.** Compared with white adipocytes, brown adipocytes are rich in mitochondria, have mitochondria with increased cristae packing, have multilocular LDs, and exhibit high protein expression of UCP1 (ref. [55]). We previously demonstrated that siRNA-mediated PLIN5 knockdown in a brown adipose cell line was associated with reduced mitochondrial oxygen consumption rate (OCR), as well as reduced mitochondrial DNA content[34]. However, is not known if PLIN5 has a direct effect on BAT mitochondria in vivo. First, we measured mitochondrial DNA as a marker of mitochondrial biogenesis. BATiPLIN5 mice had increased BAT mitochondrial DNA compared with Control mice, but only when exposed to 6 °C (Supplementary Fig. 10a). We next investigated whether PLIN5 expression affects the morphology of mitochondria by electron microscopy. Both BATiPLIN5 and Control mice were either maintained at 23 °C or exposed for 16 h to 6 °C. We measured mitochondria cristae length in individual mitochondria, and for each mitochondrion we normalized total cristae length to mitochondrion area, as a reflection of the degree of cristae packing. We observed an increase in cristae packing in mitochondria in the BAT of Control mice exposed to 6 °C, as has been previously reported[55]. Remarkably, in the BATiPLIN5 mice housed at 23 °C, we observed an increase in mitochondrial cristae packing compared to Control mice, such that the degree of mitochondrial cristae packing in the BAT of BATiPLIN5 mice housed at 23 °C resembled that of Control mice exposed to 6 °C (Fig. 6a, quantification Fig. 6b, and additional pictures in Supplementary Fig. 11).

It has been reported that PLIN5 localizes to mitochondria in muscle[30] and that PLIN5 recruits mitochondria to the LD surface[31]. The Shirihai group has reported that, during cold exposure of mice, LD–mitochondria contact sites in BAT are decreased when compared to thermoneutrality, and that mitochondria that are not in contact with ("cytoplasmic mitochondria") have increased capacity for fatty acid oxidation relative to LDs that form contact sites with mitochondria ("peridroplet mitochondria")[56]. Given these provocative findings, we quantified the number of mitochondria in contact with LDs (LD), as well mitochondria–LD contact area normalized by LD perimeter and mitochondrial perimeter (Supplementary Fig. 12a) in the BAT of BATiPLIN5 and Control mice. We observed a decrease in LD–mitochondria contact in mice exposed to 6 °C compared with 23 °C in both Control and BATiPLIN5 mice. Interestingly, we also observed a decrease in LD–mitochondria contact number and area in the BATiPLIN5 compared to Control mice, but only when housed at 23 °C. In the context of the Shirihai finding that cytoplasmic mitochondria in BAT are more oxidative than peridroplet mitochondria, our data suggest that more mitochondria in the BAT of BATiPLIN5 mice have an oxidative phenotype than in Control mice. We also measured mitochondria area and length (aspect ratio-as major axis divided by the minor axis; Supplementary Fig. 12b). We found that in the Control mice mitochondrial area was smaller when the mice were exposed to 6 °C than when maintained at 23 °C, and that

mitochondrial area in BATiPLIN5 mice at 23 °C was similar to mitochondrial area of Control mice at 6 °C.

Given the remarkable changes in mitochondrial morphology at 23 °C in the BAT of BATiPLIN5 mice compared with Control mice, we assessed mitochondrial function by two additional methods from samples obtained from mice of these genotypes housed at 23 °C. First, citrate synthase activity measured in BAT lysates was higher in BATiPLIN5 mice (Supplementary Fig. 13a). Second, we measured OCR in isolated mitochondria from mice housed at 23 °C. We decided to measure OCR in isolated mitochondria (same mass of mitochondria as determined by total protein for both genotypes) rather than intact cells, in order to eliminate the possible confounding factor of increased mitochondrial biogenesis in the BATiPLIN5 mice. Substrate-driven mitochondrial respiration in BATiPLIN5 mice was greater than in Control mice (Fig. 6c). We used guanosine diphosphate (GDP), an inhibitor of UCP1, to query the contribution of UCP1-driven respiration. Interestingly, in response to GDP the OCR of mitochondria from BATiPLIN5 mice fell to levels even lower than that of Control mice (Supplementary Fig. 13b). These data suggest that the OCR increase in the BATiPLIN5 mice is dependent on UCP1 function, or at least on intact mitochondrial function.

To further study whether UCP1 function is necessary for the phenotype in the BATiPLIN5 mice, we generated a mouse model with both inducible PLIN5 overexpression in BAT and constitutive UCP1 deficiency in BAT by crossing our BATiPLIN5 mouse with the UCP1 knockout mouse generated by Leslie P. Kozak[57] [Jackson Laboratories (B6.129-*UCP1*^tm1Kz/J Stock No: 003124)]. First, we validated the mouse models by assaying PLIN5 and UCP1 protein expression in BAT and found a 6–7-fold increase in PLIN5 protein expression in the BATiPLIN5 and UCP1KO/BATiPLIN5 mice, and no UCP1 expression in the UCP1KO and UCP1KO–BATiPLIN5 mice (Supplementary Fig. 14a). Next, we tested glucose tolerance by OGTT and found an improvement in glucose tolerance in the BATiPLIN5 mice compared with Control mice, as we observed before. However, in the UCP1KO–BATiPLIN5 mice, this improvement was not observed (Fig. 6d). In agreement with the results reported above, BATiPLIN5 mice showed smaller inguinal white adipocytes compared with Control mice (Supplementary Fig. 15a, bottom left panel); however, this feature was absent in the UCP1KO/BATiPLIN5 mice (Supplementary Fig. 15a, bottom right panel).

**Perilipin 5 associated changes in mitochondrial cristae packing require conserved ATGL and SIRT1 function.** As noted above, previous reports have suggested a role for PLIN5 in lipolysis regulation via interaction with ATGL and ABHD5[28,29]. We and others have demonstrated that PLIN5 promotes PGC1α function by disinhibiting SIRT1 deacetylase activity[34,58], and the Mashek group has elegantly shown that one mechanism of that activation is the delivery by PLIN5 of a fatty acid released by lipolysis to SIRT1 (ref. [58]). For this reason, we decided to test whether the changes in mitochondria cristae are maintained in BATiPLIN5 mice when either ATGL or SIRT1 are pharmacologically inhibited. We injected an ATGL inhibitor (Atglistatine) or Vehicle or SIRT1 inhibitor (EX-527) or Vehicle intraperitoneally daily for 7 days in BATiPLIN5 and Control mice. To confirm inhibition of ATGL by Atglistatine in BAT, we measured ex vivo lipolysis in minced BAT explants. We observed a statistically significant decrease of glycerol release from explants of both the BATiPLIN5 and Control mice with Atglistatine (Supplementary Fig. 16a). To test the inhibitory effect of EX-527 on SIRT1 deacetylase activity, we measured this activity in BAT nuclear extracts. We observed a

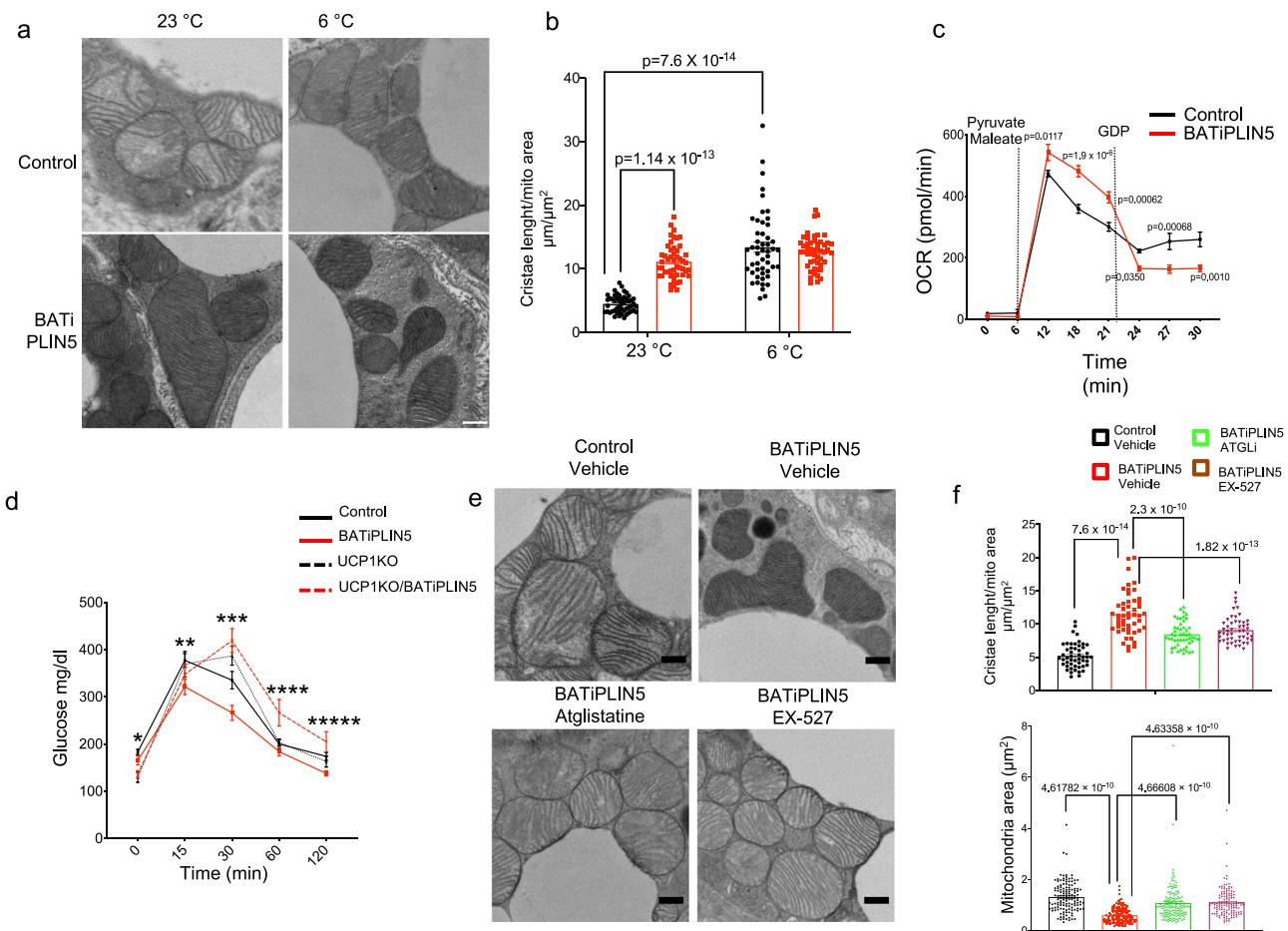

**Fig. 6 PLIN5 overexpression in brown adipose tissue increases mitochondrial cristae packing and UCP1-dependent mitochondrial respiratory function.**
**a** Representative electron microscopy (EM) images of BAT from Control or BATiPLIN5 mice housed at 23 °C or exposed at 6 °C for 16 h. Scale bar = 0.5 µm. **b** Quantification of total mitochondria cristae length normalized to mitochondrial area in BAT from Control or BATiPLIN5 mice housed at 23 °C or exposed at 6 °C for 16 h. $n = 50$ mitochondria/group. **c** Oxygen consumption rate (OCR) from BAT mitochondria isolated from Control or BATiPLIN5 mice housed at 23 °C after sequential injection of pyruvate/maleate and GDP. $n = 3$ mice per group. **d** Oral glucose tolerance test comparing Control ($n = 14$), BATiPLIN5 ($n = 11$), UCP1KO ($n = 6$), and UCP1KO/BATiPLIN5 mice ($n = 8$). **e** Representative electron microscopy (EM) images of BAT from Control or BATiPLIN5 mice housed at 23 °C and treated with Vehicle or Atglistatine or Ex-527. **f** Quantification of total mitochondria cristae length normalized to mitochondrial area in BAT from Control or BATiPLIN5 mice housed at 23 °C and treated with Vehicle or Atglistatine or Ex-527. $n = 50$. Values are mean ± s.e.m. For **b**, statistical analysis was performed using two-way ANOVA followed by Tukey posttest. P values are shown in figure. For **c**, statistical analysis was performed using unpaired two-sided Student's $t$ test. For **d**, statistical analysis was performed using two-way ANOVA followed by Tukey posttest. P values as follows, *time 0: Control vs UCP1KO $p = 0.0217$, Control vs. UCP1KO/BATiPLIN5 $p = 0.0276$, **time 15: Control vs. BATiPLIN5 $p = 0.0037$, BATiPLIN5 vs. UCP1KO $p = 0.0472$, ***time 30: Control vs. BATiPLIN5 $p = 0.0003$, Control vs. UCP1KO $p = 0.0261$, Control vs. UCP1KO/BATiPLIN5 $p = 0.00009$, BATiPLIN5 vs. UCP1KO $p = 0.000001$, BATiPLIN5 vs. UCP1KO/BATiPLIN5 $p = 0.00000000006$, ****time 60: Control vs. UCP1KO/BATiPLIN5 $p = 0.0019$, BATiPLIN5 vs. UCP1KO/BATiPLIN5 $p = 0.0002$, UCP1KO vs. UCP1KO/BATiPLIN5 $p = 0.0148$, *****time 120: BATiPLIN5 vs. UCP1KO/BATiPLI5 $p = 0.0026$. For **f**, statistical analysis was performed using one-way ANOVA followed by Tukey posttest. P values are shown in figure. Source data are provided as a Source data file.

statistically significant reduction of SIRT1 activity in nuclear extracts from both Control and BATiPLIN5 mice with EX-527 treatment (Supplementary Fig. 16b). We also observed a statistically significant increase in SIRT1 activity in BAT nuclear extracts from BATiPLIN5 mice treated with Vehicle relative to Control mice.

After confirming inhibition of ATGL and SIRT1, we performed EM in mice housed at 23 °C. As before, we observed an increase in mitochondria cristae packing and smaller mitochondria in the BATiPLIN5 compared with Control mice, but this increase was attenuated when we inhibited either ATGL or SIRT1 (Fig. 6e). Quantification of cristae length showed a statistically significant increase in the BATiPLIN5 mice treated with Vehicle that was reduced in the BATiPLIN5 mice treated with Atglistatine or EX-

527 (Fig. 6f, top panel), and mitochondrial area was increased in BATiPLIN5 mice treated with Atglistatine or EX-527 (Fig. 6f, bottom panel). Additional electron micrographs at lower magnification from this experiment are shown in Supplementary Fig. 17.

**Plin5 gene knockout in BAT impairs BAT mitochondrial respiration but leads to iWAT compensation.** To address whether expression of PLIN5 in BAT is required for maintenance of mitochondrial function in BAT and of systemic glucose homeostasis, we generated mice with constitutive PLIN5 deficiency, specifically in BAT. First, we introduced LoxP sites flanking exons 3 through 8 of the *Plin5* gene by homologous

recombination in C57BL/6 embryonic stem (ES) cells to create the *Plin5*^loxp/loxp^ strain. This strategy was chosen to minimize possible creation of an in-frame hypomorphic allele upon Cre-mediated recombination. We then crossed the *Plin5*^loxp/loxp^ mice with UCP1-Cre mice generated by Evan D. Rosen [Jackson Laboratories, B6.FVB-Tg(UCP1-Cre)1Evdr/J Stock No: 024670], as outlined in Fig. 7a. We named this model BKOPLIN5. To validate our model, we performed western blot (WB) for PLIN5 in the indicated tissues and found PLIN5 deletion only in BAT (Fig. 7b), and we confirmed these results by qPCR (Fig. 7c). We used qPCR for PLIN5 in iWAT due to the very low levels of PLIN5 protein in iWAT of wild-type mice. We found no deletion of PLIN5 in the iWAT of BKOPLIN5 mice housed at 23 °C or

exposed to 6 °C overnight (Fig. 7j). We next investigated histo-logical changes in the BAT of BKOPLIN5 mice and found a marked reduction of LDs in the BKOPLIN5 mice housed at 23 °C, exposed to 30 °C for 7 days or housed at 6 °C overnight (Fig. 7d and Supplementary Fig. 18a). We also tested BAT expression of two key thermogenic genes, *Ucp1* and *Ppargc1a*, and found a significant reduction in BKOPLIN5 mice relative to Control mice housed at 23 °C or exposed at 6 °C overnight (Fig. 7c). We per-formed electron microscopy of BAT to investigate mitochondrial morphology in the BKOPLIN5 mice at both 23 °C and overnight 6 °C. We found no marked differences in mice housed at 23 °C between Control and BKOPLIN5 mice. However, at 6 °C, while Control mice showed an increase in mitochondrial cristae

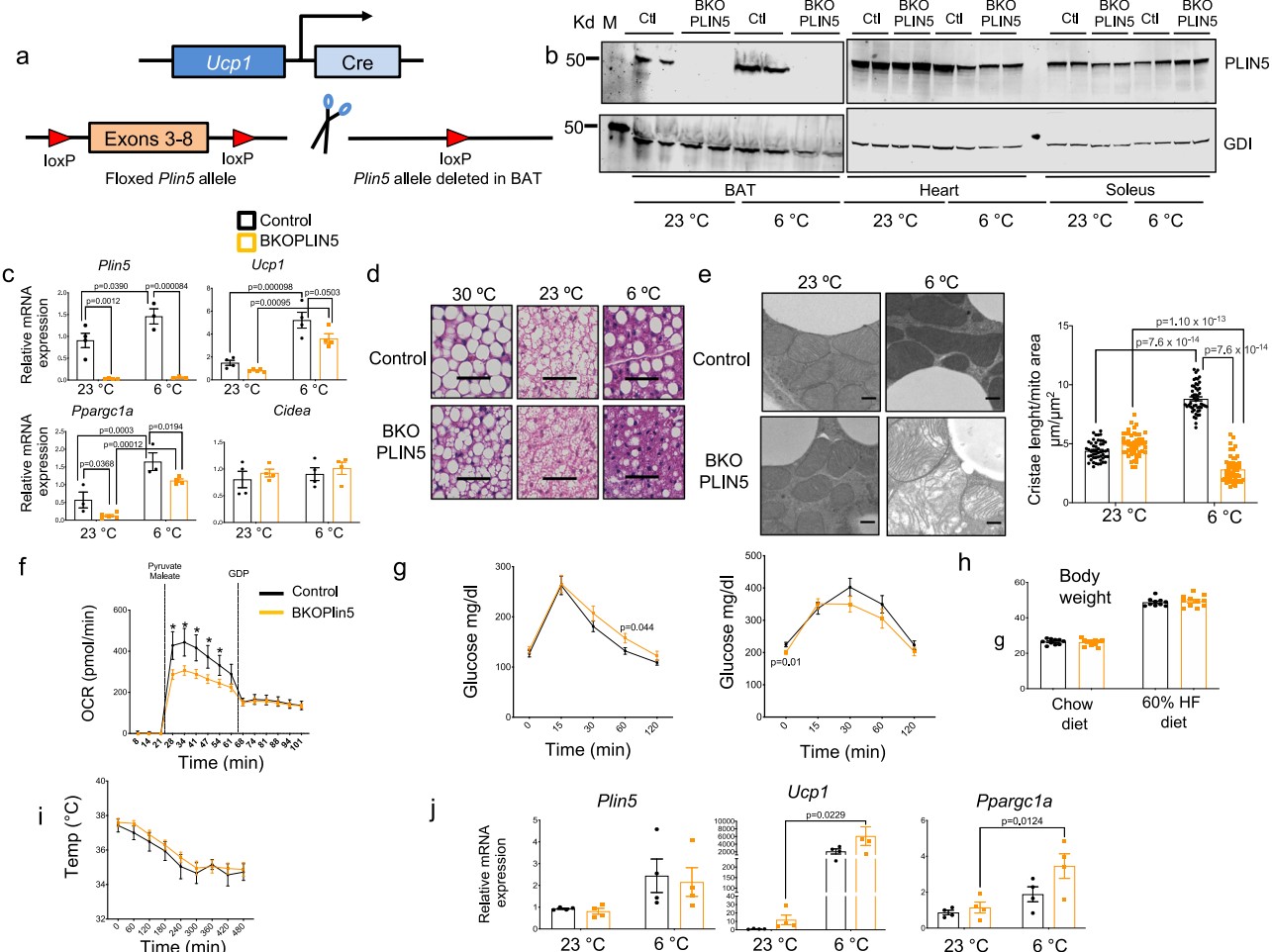

**Fig. 7 BAT-specific knockout of *Plin5* impairs BAT mitochondrial morphology and respiration but leads to iWAT compensation (data for Control mice are in black and for BKOPLIN5 mice in orange). a** Schematic representation of *Plin5* gene knockout in BAT (BKOPLIN5 mice). **b** WB for PLIN5 expression from the indicated tissues of Control or BKOPLIN5 mice housed at 23 °C or exposed to 6 °C for 16 h. **c** Relative mRNA expression by qPCR for the indicated genes in BAT from Control or BKOPLIN5 mice housed at 23 °C or exposed to 6 °C for 16 h. $n = 4$ mice per group. **d** Representative image of hematoxylin and eosin staining of BAT from Control or BKOPLIN5 from mice housed at 23 °C, housed for 7 days at 30 °C or exposed to 6 °C for 16 h. Scale bar = 100 μm. **e** Left panel: representative electron microscopy images of BAT from Control or BKOPLIN5 mice housed at 23 °C or exposed to 6 °C for 16 h. Scale bar = 0.5 μm. Right panel: quantification of total mitochondria cristae length normalized to mitochondrial area in BAT from Control or BKOPLIN5 mice housed at 23 °C or exposed to 6 °C for 16 h. $n = 50$ mitochondria/group. **f** Oxygen consumption rate (OCR) from BAT mitochondria isolated from Control or BKOPLIN5 mice housed at 23 °C after sequential injection of pyruvate/maleate and GDP. $n = 3$ per group. **g** OGTT after 8 weeks on chow diet (left panel) or HF diet (right panel). Chow diet Control $n = 10$, BKOPLIN5 $n = 11$. **h** Weight of Control ($n = 10$ per group) or BKOPLIN5 ($n = 11$ per group) mice fed chow or HF diet for 12 weeks. **i** Acute cold tolerance test performed on Control ($n = 5$ mice) or BKOPLIN5 ($n = 7$ mice). **j** Relative mRNA expression by qPCR for the indicated genes in iWAT from Control or BKOPLIN5 mice housed at 23 °C or exposed 6 °C for 16 h. $n = 4$ mice per group. Values are mean ± s.e.m. For **c**, **e**, and **j**, statistical analysis was performed using two-way ANOVA followed by Tukey posttest. *P* values are shown in the figure. For **f** and **g**, statistical analysis was performed using unpaired two-sided Student's *t* test. For **g** and **h**, *p* values are shown in the figure. For **f**, *p* values as follow: *time 28 $p = 0.0016$, *time 34 $p = 0.0021$, *time 41 $p = 0.0044$, *time 47 $p = 0.0141$, *time 54 $p = 0.0473$. Source data are provided as a Source data file.

packing compared with 23 °C, BKOPLIN5 mice instead had a decrease in cristae packing and altered mitochondrial cristae morphology at 6 °C (Fig. 7e, left panel and Supplementary Fig. 19a). As a quantitative measure of cristae packing, we assessed total cristae length normalized to mitochondrial area. By this measure, Control mice had increased total cristae packing when housed at 6 °C compared with 23 °C, but BKOPLIN5 mice had a marked decrease at 6 °C compared with 23 °C (Fig. 7e, right panel). We found no difference between Control and BKOPLIN5 mice in LD–mitochondria contacts (Supplementary Fig. 19b). Mitochondrial area was increased and mitochondrial length decreased in the BKOPLIN5 mice during cold exposure in BKOPLIN5 compared with Control mice (Supplementary Fig. 20a).

To test mitochondrial function in the BKOPLIN5 mice, we measured OCR in isolated mitochondria obtained from mice housed at 23 °C. We observed a significant reduction in OCR in BKOPLIN5 mice compared with Control mice (Fig. 7f). We next investigated the systemic effects of BAT PLIN5 deficiency. We found a small decrease in glucose tolerance at the 60 min timepoint in BKOPLIN5 mice on chow and no difference in glucose tolerance on HFD (Fig. 7g). There were no differences in weight between genotypes (Fig. 7h). We also performed acute cold tolerance testing and found no differences between genotypes (Fig. 7i). This was a surprising result given the improved cold tolerance by 8 h that we observed with PLIN5 gain-of-function in BAT reported above, so we examined whether there were compensatory secondary effects on iWAT in BKOPLIN5 mice. We found statistically significant increased Ucp1 and Ppargc1 gene expression in iWAT after 16 h at 6 °C in the BKOPLIN5, but only a trend in the Control mice. There were no changes in the iWAT Plin5 gene expression in either genotype (Fig. 7j). We found no differences in adipocyte size or appearance in H&E staining of iWAT (Supplementary Fig. 21a). These data suggest that changes in iWAT gene expression to promote beiging may compensate for BAT PLIN5 deficiency in terms of acute cold tolerance.

## Discussion

In this study, we provide the first report that in wild-type mice PLIN5 mRNA and protein expression levels markedly increase in BAT during exposure to cold. Based on the data from genetically engineered mouse strains, we demonstrate herein that PLIN5 in BAT plays a critical role in the adaptive response of BAT to cold stress with respect to fatty acid uptake and oxidation, increased cristae packing in mitochondria, and UCP1-dependent mitochondrial respiration. Further, we show that PLIN5 function in BAT influences systemic glucose metabolism via secondary effects on iWAT (adipocyte size, thermogenic gene expression, inflammatory gene expression, insulin signaling, and lipolysis) and liver (steatosis, insulin signaling).

In our hands, driving increased expression of PLIN5 in brown adipocytes using the Tet-On system in mice housed at 23 °C mimicked the effect of overnight exposure to 6 °C, with respect to cristae packing in mitochondria. Because mitochondrial cristae are the subcompartment of the mitochondrial inner membrane that harbors the machinery of oxidative phosphorylation[59], it is not surprising that isolated mitochondria from BATiPLIN5 mice have augmented substrate-driven respiration. Augmentation of mitochondrial cristae, a trend toward increased mitochondrial DNA, increased citrate synthase activity, and increased respiration were consistent with the PLIN5-dependent changes in BAT thermogenic gene expression that we observed at 23 °C, and even more so at 6 °C. In line with our previous report that nuclear PLIN5 activates SIRT1 activity[34], we observed an increase in

SIRT1 activity in BATiPLIN5 mice. We also demonstrated that SIRT1 inhibition attenuates the increase in mitochondrial cristae packing that we have described in the BATiPLIN5 mice. The effects of PLIN5 on gene expression in BAT were consistent with our previous published report that during catecholamine-stimulated lipolysis nuclear PLIN5 promotes transcription of genes involved in mitochondrial biogenesis and function via interaction with SIRT1–PGC1α complexes[34].

In keeping with the augmentation of both thermogenic gene expression and of mitochondrial function in BAT, BATiPLIN5 mice were more tolerant to acute cold challenge by 8 h of fasting than Control mice, and they maintained a higher core body temperature throughout 3 weeks of cold acclimation when fed ab libitum. Two elegant studies in genetically engineered mouse models have demonstrated that adaptive non-shivering thermogenesis in BAT requires fatty acids derived from triglyceride lipolysis in WAT during fasting or from circulating triglyceride-rich lipoproteins in the fed state to serve as substrates for uncoupled mitochondrial respiration and to activate UCP1 (refs. [60,61]). Surprisingly, lipolysis in brown adipocytes is not required for adaptive thermogenesis in BAT[60,61]. In BATiPLIN5 mice, neither basal nor β-3 agonist-stimulated lipolysis was augmented compared with Control mice, as assessed by measurement of glycerol efflux from minced BAT tissues. In other experimental contexts, PLIN5 has been shown to have a role in limiting basal lipolysis and augmenting catecholamine-stimulated lipolysis, both in cells[28,29] and in mouse models of PLIN5 over-expression in heart[36,62] or knockout in heart[63] and liver[38]. It is possible that PLIN5 plays a less central role in regulating lipolysis in BAT due to the high level of Perilipin 1 (PLIN1), which is also expressed in BAT[25] and is proposed to regulate lipolysis in BAT[64]. Even though BAT PLIN5 overexpression does not affect BAT lipolysis, systemic pharmacological inhibition of ATGL lipase activity attenuates the phenotype of increased cristae packing in the mitochondria of BATiPLIN5 mice. Additional studies are necessary to clarify this effect, because it remains unclear whether it is inhibition of ATGL in other tissues, such as WAT or heart, or of PLIN1-regulated ATGL activation in BAT that blocks the phenotype.

In contrast to lipolysis in BAT, β-3 agonist-stimulated lipolysis in iWAT was increased in BATiPLIN5 mice. Increased basal and catecholamine-stimulated lipolysis have been associated with white adipocyte hypertrophy[65], but in BATiPLIN5 mice inguinal white adipocytes were significantly smaller than in Control mice. Smaller adipocytes may be a marker for greater systemic insulin sensitivity. For example, in a study of healthy overweight adults, a hypocaloric diet resulted in weight loss and improvement in insulin sensitivity, and the best predictors of improvement were decreased adipocyte size and reduced waist circumference[66]. The reduction in inguinal white adipocyte size in BATiPLIN5 mice suggests a healthy remodeling of the iWAT depot, which is supported by our finding of decreased markers of inflammatory gene expression, increased insulin signaling as assessed by phospho-AKT/total AKT, and increased Glut4 mRNA. In other mouse models, Glut4 gene expression in iWAT correlates positively with systemic insulin sensitivity[67]. The smaller size of white adipocytes in the iWAT of BATiPLIN5 mice was not associated with any changes in whole body fat mass, as determined by nuclear magnetic resonance at 23 °C. This finding suggests the possibility that increased PLIN5 expression in BAT leads to the increased recruitment of new adipocytes in iWAT, an adipose depot that is resistant to adipocyte hyperplasia and generally expands by hypertrophy, especially in male mice[68]. It is important to note that, in the iWAT of BATiPLIN5 mice, gene expression for PLIN5 and other thermogenic genes is downregulated, especially during acute cold exposure, and there is a trend toward

decreased expression of these genes at normal housing temperatures. This finding may point to the existence of compensatory negative feedback from BAT to iWAT, such that increased thermogenic activity in BAT leads to the reduced thermogenic activity in iWAT. In any event, the reduction in iWAT cell size and increase in iWAT insulin sensitivity are not explained by iWAT beiging, as indicated by the reduced expression of *Ucp1* in that depot and the unchanged expression of markers of sympathetic innervation. It will be important to determine whether brown adipocytes in BATiPLIN5 mice secrete a factor or factors that stimulate WAT hyperplasia, as well as promote WAT lipolysis to fuel BAT uncoupled respiration.

In addition to fatty acids derived from lipolysis in WAT, fatty acid substrates to fuel thermogenesis in BAT may be derived from glucose uptake and de novo lipogenesis in BAT[40–42], or from the hydrolysis of triglycerides in circulating triglyceride-rich lipoproteins[24]. In BATiPLIN5 mice, we found that increased lipolysis in iWAT was accompanied by the increased uptake and oxidation of fatty acids in BAT. When housed at 23 °C, BATiPLIN5 mice did not exhibit differential expression in BAT of genes involved in the uptake of fatty acids into brown adipocytes (*Cd36*, *Fatp1*) or in the release of fatty acids from triglyceride-rich lipoproteins (*Lpl*, *Angplt4*); however, short-term exposure of these mice to 6 °C was associated with a significantly greater rise in *Lpl* RNA in BATiPLIN5 compared with Control mice. We did not observe an increase in BAT glucose uptake in the BATiPLIN5 mice, and DNL gene expression after short cold exposure was similar between Control and BATiPLIN5 mice. Interestingly, housing mice under the mild cold stimulus of 23 °C revealed an increase in DNL gene expression in BATiPLIN5 mice compared with Control mice. In a recent study, mice exposed to mild cold (22 °C) for 28 days showed an induction of DNL genes, when compared with mice housed at thermoneutrality[41]. These collective data may delineate a dual role for PLIN5 during cold adaptation, whereby PLIN5 is involved in promoting[1] de novo lipogenesis during chronic mild chronic cold exposure, and[2] LPL-mediated fatty acid uptake during more severe acute cold exposure.

The healthy remodeling of iWAT in BATiPLIN5 mice was accompanied by reduced hepatic TAG content on chow and resistance to hepatic steatosis on a HFD, which we documented by histology and measurement of hepatic triglycerides. BATiPLIN5 mice also exhibited reduced blood glucose excursions during pyruvate tolerance testing, which is consistent with a reduction in hepatic gluconeogenesis. The reduction in both hepatic steatosis and hepatic gluconeogenesis would follow from improved iWAT insulin sensitivity and hepatic insulin sensitivity, respectively, which we demonstrated by the assessment of insulin-stimulated AKT phosphorylation in those tissues. Collectively, our data strongly support an increase in both hepatic and iWAT insulin sensitivity. Ultimately, in BATiPLIN5 mice, the improved insulin sensitivity in liver and iWAT was associated with improved glucose tolerance achieved at lower circulating insulin levels, which suggests improvement in systemic insulin sensitivity.

Notably, the effects of PLIN5 overexpression in BAT on systemic glucose tolerance were negated when BATiPLIN5 mice were bred onto a UCP1 knockout background, and the decrease in iWAT adipocyte size was also lost. Also, during mitochondrial respiration experiments, the mitochondrial OCR of BATiPLIN5 mice showed a greater inhibition to GDP (a UCP1 inhibitor) than that of Control mice. A study previously reported that UCP1 knockout on the C57BL/6 background strain is associated with reduced electron transport chain proteins in mitochondria, as well as dysmorphic mitochondria during chronic cold acclimation[69]. Thus, the positive metabolic effects of increased

PLIN5 expression in BAT are dependent on UCP1 expression and/or intact mitochondrial respiratory function. Remarkably, PLIN5 BAT overexpression has a profound effect on increasing mitochondrial cristae packaging even at 23 °C, and the BKO-PLIN5 mice showed loss of mitochondrial cristae during cold exposure.

PLIN5 knockout specifically in BAT (BKOPLIN5) was not associated with marked worsening of glucose tolerance, as might have been predicted by the improvement of glucose tolerance observed with PLIN5 overexpression in BAT, though there was a small statistically significant worsening at the 60-min timepoint in the OGTT. Nor was cold tolerance reduced in BKOPLIN5 mice, as might have been expected given the improved cold tolerance we observed in BATiPLIN5 mice. These data raise the question of whether the glucose and cold tolerance phenotypes of the BATiPLIN5 mice reflect the normal function of PLIN5 in BAT or rather represent neomorphic phenotypes. We consider the possibility of neomorphic phenotypes unlikely, because the BAT of BKOPLIN5 mice has significantly reduced expression of key thermogenic genes, has reduced substrate-driven respiration, and exhibits reduced mitochondrial cristae packing during cold stress compared with Control mice. The mostly preserved glucose tolerance and cold tolerance in BKOPLIN5 mice are likely due to compensation that occurs in iWAT in the setting of chronic PLIN5 deficiency in BAT, as suggested by a trend toward increased *Ucp1* at 23 °C and *Ppargc1a* and *Ucp1* at 6 °C in the BKOPLIN5 iWAT compared to Control mice. Also, *Ppargc1a* and *Ucp1* gene expression in iWAT showed a more robust and statistically significant response to overnight cold exposure in the BKOPLIN5 mice, when compared to Control mice.

The idea that human brown or beige adipose tissue is a promising target for the development of therapeutics to address type 2 diabetes is reinforced by our finding that increasing PLIN5 in mouse BAT promotes glucose tolerance, healthy remodeling of iWAT, and protection against hepatic steatosis on a HFD. We have shown that more PLIN5 in brown adipocytes augments mitochondrial oxidative function, and promotes the uptake and oxidation of fatty acids in BAT. This model (Fig. 8) is consistent with the conception of BAT as a "metabolic sink" that utilizes fatty acids mobilized from WAT as fuel for uncoupled mitochondrial respiration. Further, with the additional stimulus of cold exposure, BATiPLIN5 mice lose more weight than Control mice despite eating more food. This finding suggests the possibility that pharmacologic augmentation of PLIN5 expression coupled with an additional stimulus might work synergistically to promote loss of adipose mass as a means to treat obesity. How to effectively, safely, and specifically promote PLIN5 expression in the thermogenic adipose tissue of mice and humans is not currently known. Our studies reported here suggest the potential of PLIN5 to mimic the effect of cold temperature on BAT and, thereby, to improve systemic glucose tolerance and protect against diet-induced hepatic steatosis.

Our current work has several limitations. We have not yet discovered the mechanistic link between augmented PLIN5 function in BAT and the promotion of smaller, more insulin sensitive white adipocytes in WAT, though we have shown that the mechanism is not increased beiging of iWAT nor increased secretion of FGF21 by BAT. Also, we have not shown directly that the smaller iWAT adipocytes are due to hyperplasia, which would be unusual in that depot, at least in male mice. Finally, though we think it likely that the effects of PLIN5 on mitochondrial structure and function in BAT are mediated in large part by promotion of the increased activity of the SIRT1/PGC1α transcriptional program, we have not ruled out a role for PLIN5 via its direct physical association with mitochondria. These questions remain for further study.

**Fig. 8 Working model for PLIN5 function in BAT and its role in systemic metabolism.** Increasing PLIN5 expression in mouse BAT is sufficient to promote increased fatty acid uptake, fatty acid oxidation, and mitochondrial uncoupled respiration in BAT. One mechanism for these changes is the promotion of the thermogenic gene program. The effects of PLIN5 overexpression on BAT mitochondria mimic the effects of cold exposure in terms of increased cristae packing. These changes in BAT are accompanied by improved acute cold tolerance and chronic cold acclimation and by improved systemic glucose tolerance and resistance to hepatic steatosis on high-fat diet. The improved glucose tolerance is accompanied by healthy iWAT remodeling, as reflected by reduced white adipocyte size and increased insulin sensitivity in iWAT. Genetic deletion of PLIN5 in BAT reveals that PLIN5 is required for normal mitochondrial respiratory function in BAT, as well as for maintenance and augmentation of mitochondrial cristae architecture during cold exposure. Preserved cold tolerance and glucose metabolism in the absence of BAT PLIN5 is likely due to compensatory induction of thermogenic pathways in iWAT.

## Methods

**Animal studies.** We performed all animal experiments with approval from the University of Texas Southwestern Medical Center (UTSW) Institutional Animal Care and Use Committee, and all experiments were performed in adherence to the guidelines of National Research Council, 2011, *Guide for the Care and Use of Laboratory Animals: Eighth Edition*[70].

The UTSW Transgenic Core assisted in generation of the TRE-Plin5 and Plin5$^{loxP/loxP}$ genetic mouse models on a C57BL/6 J background. For all experiments presented in this study, we used male mice on a C57BL/6J background. We housed mice in a conventional animal facility at 23 °C in a 12-h light/dark cycle with free access to food and water, unless otherwise indicated in the text, figure legends, or "Methods". For controlled temperature experiments, we housed mice in a thermoneutrality box at 30 °C (Powers Scientific Inc., Model # RIS70SD) or cold box (Powers Scientific Inc., Model # RIS70SD) at 6 °C. Mouse euthanasia was by isoflurane anesthesia followed by cervical dislocation.

### Generation of mouse lines

*BATiPLIN5 strain.* The TRE-Plin5 allele was generated by subcloning a cDNA encoding mouse PLIN5 with an in-frame amino-terminal epitope tag[25] downstream of the TRE promoter of a DOX-inducible plasmid that was kindly provided by Philipp Scherer[71]. The resulting TRE-Plin5 plasmid was then linearized and introduced by pronuclear injection into mice on albino C57BL/6J background at the UTSW transgenic core. The primer pair for genotyping the TRE-Plin5 allele was as follows: Plin5-gt1-f, 5′-CATGACTGAGGCTGAGCTAGCAG-3′; Plin5-gt1-r, 5′-CCTCTCTGCATATGCTGGATC AGC-3′. The BATiPLIN5 strain was generated by crossing the TRE-Plin5 strain with UCP1$^{rtTA}$ strain, which was a gift from Philipp Scherer[72].

*UCP1KO/BATiPLIN5 strain.* We created this mouse line by crossing UCP1KO mice generated by Leslie P. Kozak[57] and obtained from Jackson Laboratories (B6.129-*Ucp1*$^{tm1Kz}$/J Stock # 003124) with BATiPLIN5 mice.

*BKOPLIN5 strain.* To create a conditional *Plin5* allele in mice, two loxP sites were introduced flanking exons 3–8 of the *Plin5* gene (NM_025874.3). An FRT-PKG-Neo-FRT cassette[73] followed the loxP site flanking exon 8, which generated a knockout-first allele. BAC injection of the targeting construct and homologous recombination in C57BL/6J ES cells was performed by the UTSW Transgenic Core. The correct ES cell

clones were screened and verified by Southern blot. The founder was backcrossed to the C57BL/6J strain. The knockout-first allele was crossed with flp mice (JAX 009086) in order to remove the Neo cassette and generate the floxed line, Plin5$^{loxP/loxP}$. The final knockout allele (deletion of exons 3–8) was generated by crossing Plin5$^{loxP/loxP}$ mice with UCP1-Cre mice to generate the BKOPLIN5 strain. The UCP1-Cre mouse line (B6.FVB-Tg(Ucp1-Cre)1Evdr/J) was generated by the Evan Rosen Lab[74] and obtained from Jackson Laboratories Stock # 024670. The primer pair for the knockout-first allele was 5′-CTGATGCTCTTCGTCCAGATC-3′, 5′-GTGCTCGACG TTGTCACTGA-3′. The primer pair for the floxed and wild-type alleles is Plin5-5loxp-f: 5′-GAACTCATCCTCGTCCCACC-3′; Plin5-5loxp-r: 5′-CCTGAGCTGTC TGATCACCG-3′; WT 212 bp, flox 310 bp. The primer pair for the knockout allele is 5′-CTCACCAGGTCATTCCCTCTA-3′, 5′-GGCTTGCTTCAGTTTGCCAT-3′. We confirmed the genotypes for all mice used in experiments by PCR. The genotypes of the Control and Experimental mice strains used in this manuscript are as follows: For BATiPLIN5: Control strain UCP1rtTA$^{+/+}$; TRE-Plin5$^{-/-}$, Experimental strain UCP1rtTA$^{+/+}$; TRE-Plin5$^{+/-}$. For UCP1KO Background: Control strain UCP1rtTA$^{+/+}$; TRE-Plin5$^{-/-}$; UCP1KO$^{-/-}$, Experimental strain 1 (UCP1KO) UCP1KO-UCP1rtTA$^{+/+}$; TRE-Plin5$^{-/-}$; UCP1KO$^{+/+}$, Experimental strain 2 (BATiPLIN5) UCP1rtTA$^{+/+}$; TRE-Plin5$^{+/-}$; UCP1KO$^{-/-}$, Experimental strain 3 (UCP1KO/BATiPLIN5) UCP1rtTA$^{+/+}$; TRE-Plin5$^{+/-}$; UCP1KO$^{+/+}$, For BKOPLIN5: Control strain Plin5 $^{loxp/loxp}$; Ucp1-Cre$^{-/-}$; Experimental strain-Plin5 $^{loxp/loxp}$; Ucp1-Cre$^{+/-}$.

BATiPLIN5 and BKOPLIN5 mice are available upon request from the corresponding author of this manuscript.

Other mouse lines used in this manuscript can be obtained from standard commercial sources specified in these methods.

**Diets and timeline for experiments.** For BATiPLIN5 and UCP1KO/BATiPLIN5 mice, we used special diets that contained 200 mg of DOX/kg diet and are referred to as Chow (Bioserve, S3888) or HFD (60% high fat, Research Diets, D09050201). Unless otherwise indicated, we started the special diet 8 weeks after birth, and in general experiments were performed 12–16 weeks after birth (4–8 weeks after initiation of special diet). For all experiments involving the DOX-inducible transgene (TRE-Plin5), DOX-containing diets were fed to both Control and experimental groups. For the BKOPLIN5 mice, we used chow diet (Teklad, 2016) or 60% HFD (Research Diets, D12492) without DOX. Unless otherwise indicated, we started HFD at 8 weeks after birth, and experiments were performed 8 weeks after initiation of the special diet.

**Cold/thermoneutrality experiments**. For Fig. 1b, c, we transferred mice from 23 °C to the indicated temperature box (30 or 6 °C), and after 16 h, we harvested BAT for protein and RNA extraction.

For Fig. 1e, we transferred mice from 23 °C to the indicated temperature box (30 or 6 °C), and harvested BAT for protein and RNA extraction at the indicated time points. For body temperature measurements, we used an implantable temperature transponder (IPTT300, BioMedic Data Systems Inc, Seaford, DE) inserted subcutaneously in the back of the mice, using the manufacture's needle assembly under general anesthesia with isoflurane via precision vaporizer. To allow recovery, we performed temperature experiments 2–3 days after the transponder insertion. For temperature experiments, we single-housed mice and measured temperature using a temperature reader (DAS-8007-IUS, BioMedic Data Systems Inc, Seaford DE), as described below. For Fig. 3b experiment, we measured temperature before we transferred the mice from 23 °C to the 6 °C temperature box. At the time of transfer, we removed food and fasted the mice for 8 h. We measured body temperature, as described above, every hour for 8 h. After 8 h, we returned food to the mice and then maintained the mice in the cold box. For the experiment in Fig. 1c, we measured temperature in mice from experiment described above at 9 a.m. on the indicated days, and provided access to food ad libitum. For the experiment in Fig. 3d, we maintained mice from the experiment described above for 21 days in the cold box. At day 22, food was withdrawn, and we measured body temperature of the mice at the indicated time points.

**Body weight**. We performed all body weight measurements between 9 and 10 a.m. in fed mice.

**Food intake**. For food intake measurement, we single-housed mice and measured food weight every morning at 9 a.m. for at least 6 days.

**Energy expenditure**. Food intake, meal patterns, energy expenditure, and loco-motor activity were monitored at the UTSW Metabolic Phenotyping Core, using a combined indirect calorimetry system (Labmaster, TSE Systems GmbH, Germany). Mice were individually housed in a light (12 h on/12 h off, 7 a.m.–7 p.m.) and temperature (22.5–23.5 °C) controlled environment and acclimated in the home cage for 5 days before data collection. We analyzed mice in the metabolic chambers for 4 days with food and water ad libitum. We measured $O_2$ consumption and $CO_2$ production by indirect calorimetry to determine the energy expenditure. We measured locomotor activity by multidimensional infrared light beam detection system. We recorded continuous food and water intake using lid-mounted sensors. We normalized data to lean body mass as determined, using a Bruker MQ10 NMR analyzer.

**Body composition**. We measured body fat mass and lean mass in conscious mice, using the Bruker Minispec mq10 NMR (UTSW Metabolic Phenotyping Core).

**OGTT and ITT**. For OGTT, we fasted mice for 5 h. We administered 2.5 g of glucose/kg of body weight by oral gavage and collected tail blood at the indicated time points for measurement of glucose and insulin. For glucose measurement, we used a Contour Next EZ glucometer (Bayer HealthCare LLC), and for insulin, we used a commercially available ultrasensitive mouse insulin ELISA kit (Crystal Chem Inc, Cat # 90080), and followed the manufacturer's instructions. For ITT, we fasted mice for 5 h. We administered insulin Humulin R U-100 (Lilly USA, Cat # HI213) by intraperitoneal injection (1 unit of insulin/kg body weight). We collected tail blood at the indicated time points and measured glucose using the Bayer glucometer.

**Serum triglycerides and NEFA measurements**. For triglycerides and NEFA measurements, we collected blood for serum preparation after overnight fasting (Fig. 4d) or fed after overnight exposure to different temperatures (Fig. 4e). For triglycerides measurement, we used Infinity™ triglycerides liquid stable reagent (Cat # TR22421, Thermo Scientific). For calibration, we used Multi calibrator lipids (Cat # 464-01601, Wako). For NEFA quantification, we used HR series NEFA-HR (Cat # 999-34691, Wako) with NEFA standard solution (Cat # 276-76491, Wako).

**Histology and adipocyte size quantification and Oil red O staining**. To obtain mice tissue samples for histology, we performed cardiac perfusion under ketamine anesthesia. After cardiac perfusion with 4% paraformaldehyde in phosphate-buffered saline (PBS), pH 7.4, we dissected tissues and fixed with 4% paraformaldehyde solution overnight. The UTSW Molecular Pathology Core Facility performed paraffin embedding, sectioning, H&E staining, and Oil red O staining. We acquired bright-field images using a Keyence BZ-X710 microscope. Keyence BZ-X Analyzer software was used for analysis of bright-field images of H&E-stained paraffin sections. For iWAT adipocyte size quantification, >200 adipocytes were quantified in each individual animal ($n = 3$ mice per group).

**Pyruvate tolerance test**. We overnight fasted mice and then intraperitoneally injected them with 1 g/kg body weight pyruvate (Cat # P5280 Sigma- Aldrich) diluted in PBS, pH 7.3. We collected tail blood immediately before injection and after injection at the time points indicated in the figure, and measured glucose with the Bayer glucometer referenced above.

**Citrate synthase activity**. We used a citrate synthase activity colorimetric assay kit (Cat # 318-100, BioVision), and followed the manufacturer's instructions. We homogenized BAT (~10 mg) with 100 μl of the provided citrate synthase assay buffer, we centrifuged the homogenate for 5 min at 10,000 × g in a microfuge, and collected the supernatant. We added 10 μl of each sample into wells of a 96-well plate and adjusted the volume of each sample to 50 μl with assay buffer. For standard curve preparation, we used the standards provided with the kit. We added reaction mix provided with the kit and immediately read absorbance (OD 420 nm) in a kinetic mode at 25 °C for 40 min.

**Mitochondria Isolation and OCR measurements**. We resuspended ~50 mg of minced BAT in 5 ml of isolation buffer containing 70 mM sucrose, 210 mM mannitol, 5 mM HEPES, 1 mM EGTA, and 0.5% (w/v) fatty acid-free bovine serum albumin (BSA, pH 7.2) and homogenized the sample with a motorized Dounce homogenizer by 30 strokes at 500 r.p.m. We centrifuged the homogenate at 800 × g for 10 min at 4 °C and collected the supernatant. We centrifugated the supernatant at 8000 × g for 10 min at 4 °C and discarded the supernatant. We washed the resulting pellet twice with isolation buffer. We resuspended final pellet in 200 μl of resuspension buffer containing 70 mM sucrose, 220 mM mannitol, 10 mM $KH_2PO_4$, 5 mM $MgCl_2$, 2 mM HEPES, 1 mM EGTA, and 0.2% (w/v) fatty acid-free BSA. We measured protein concentration with Coomassie Bradford protein assay kit (Thermo Scientific, Cat # 23200). We measured OCRs of the isolated mito-chondria using a Seahorse XFp Analyzer (Agilent). Specifically, we loaded 10 μg of isolated mitochondria in 80 μl of resuspension buffer in an XFp miniplate and sequentially injected pyruvate (final concentration 10 mM) and maleate (final concentration 5 mM) and GDP (final concentration 1 mM), and measured OCR at the time points indicated in the figures.

**Mitochondrial DNA quantification**. For isolation of total DNA, 25 mg of tissue was homogenized in PBS using a TissueLyser II and stainless-steel beads (Qiagen, 69989). Samples were processed with Qiagen's QIAamp DNA Mini Kit (51304) per the kit instructions. Mitochondrial DNA was amplified using primers specific for the mitochondrial cytochrome $c$ oxidase subunit 2 (COX2) gene, and normalized to genomic DNA by amplification of the ribosomal protein s18 (rps18) nuclear gene, using quantitative PCR. We used primers listed in primers table below that were previously described[75].

**EX-527 administration and SIRT1 activity assay**. For measurement of SIRT1 deacetylase activity, we used SIRT1 activity assay kit (Abcam Cat # ab156065) following the manufacturer's instructions. We measured SIRT1 activity from nuclear extracts from BAT of BATiPLIN5 or Control mice treated with SIRT1 inhibitor EX-527 or Vehicle (Millipore Sigma E7034) at 10 mg/kg/day by intra-peritoneal injection daily for 7 days along with Dox diet. After the treatment, we dissected BAT and isolate nuclear extracts as described below. For each reaction and following the volumes recommended by the manufactures instructions, we added to microplate wells 25–30 μl ddH₂O, 5 μl of SIRT1 assay buffer, 5 μl fluoro-susbtrate peptide, 5 μl NAD, and 5 μl of developer. To initiate the reactions, we added our test sample with a final protein concentration of 5 μg in 5 μl volume. For the experiment, we included the following controls: blank control (nuclear extraction buffer), positive control (recombinant SIRT1), and no NAD control. After initiation of the reaction, fluorescence intensity was read at 340–360 nM excitation and 440–460 nM emission. After 1-h reaction was stopped with Stop solution and activity was calculated as the differences in fluorescence intensity between test sample and blank control.

**Nuclear isolation for SIRT1 activity assay**. We performed all the steps of nuclear isolation at 4 °C. We collected ~100 mg of BAT and minced the tissue into small pieces. We resuspended the BAT in 2 ml of isolation buffer containing 10 mM Tris-HCl (pH 7.5), 10 mM NaCl, 15 mM $MgCl_2$, 250 mM sucrose, and 0.1 mM EGTA. Tissue was homogenized with a Dounce homogenizer (40–50 strokes) and the homogenate was decanted into a centrifuge tube, maintained on ice for 30 min, vortexed at maximum speed for 15 s, and then centrifuged at 800 × g for 15 min. After centrifugation, the supernatant was removed and the pellet was resuspended in isolation buffer and centrifuged at 500 × g for 15 min, and subsequently washed in isolation buffer one additional time followed by centrifugation at 1000 × g for 15 min. Finally, the pellet was resuspended in 50 μl of extraction buffer containing 50 mM HEPES KOH (pH 7.5), 420 mM NaCl, 0.5 mM EDTA, 0.1 mM EGTA, and 10% glycerol, sonicated for 30 s, and maintained on ice for 30 min. Finally, the sample was centrifuged at 20,000 × g for 10 min, and the supernatant was used for SIRT1 activity assay.

**Electron microscopy**. For transmission electron microscopy sample collection, we performed cardiac perfusion under ketamine anesthesia with a perfusion buffer (4% paraformaldehyde, 1% glutaraldehyde, and 0.1 M sodium cacodylate, pH 7.4), and we dissected BAT into 1 mm pieces that were then fixed with 2.5% glutaraldehyde and 0.1 M sodium cacodylate, pH 7.4. Further processing of the samples was performed at UTSW Electron Microscopy Core as follows: tissue samples were rinsed in 0.1 M sodium cacodylate buffer and postfixed in 1% osmium tetroxide and 0.8% potassium ferricyanide in 0.1 M sodium cacodylate buffer three times for 3 h at room temperature. After three rinses in water they were stained en bloc with 4% uranyl acetate in 50% ethanol for 2 h. Next, the samples were dehydrated with increasing concentrations of ethanol, transitioned into resin with propylene oxide, infiltrated with Embed-812 resin, and polymerized in a 60 °C oven overnight. Blocks were sectioned with a diamond knife (Diatome) on a Leica Ultracut 7 ultramicrotome (Leica Microsystems) and collected onto copper grids, post stained with 2% aqueous uranyl acetate and lead citrate. Images were acquired on a JEOL 1400 Plus electron microscope and photographed with a BIOSPR16 camera.

**Electron micrograph image quantification**. For quantification of LD–mitochondria contact sites, mitochondria length, and aspect ratio, we used Image J software NIH Ver 1.53 and followed protocol in ref. [56]. We quantified mitochondria in contact with LDs by count and contact area as % of mitochondrial perimeter or % of LD perimeter ($n = 10$ EM per group for LD–mitochondria contact) and for mitochondria length and area (all the mitochondria in five EM fields per group). For cristae quantification, we measured the total length of cristae in each mitochondrion and normalized by mitochondria area ($n = 50$ mitochondria per group).

**Insulin signaling by measurement of AKT and pAKT**. Mice were intraperitoneally injected with saline or insulin 1.5 U/kg body weight. Mice were sacrificed at the indicated time points after injection, and tissues (iWAT or liver) were immediately dissected and prepared for western blotting to assess AKT and pAKT (S473).

**Ex vivo lipolysis assay, glycerol measurement, and Atglistatine administration**. We measured ex vivo lipolysis in BAT and iWAT tissues explants as described before[76], with minor modifications. We collected BAT and iWAT explants (~20 mg) from fed Control or BATiPLIN5 mice. We minced the fat explants, suspended them in incubation media (DMEM supplemented with 2% fatty acid-free BSA), and transferred the minced explants into wells of a 96-well plate. For basal lipolysis, minced fat explants were incubated for 60 min in 150 µl of fresh incubation media, after which we measured glycerol in the incubation media. For stimulated lipolysis, we preincubated minced explants for 60 min in 150 µl incubation media sup1-plemented with 10 µm of CL-316,243 (Sigma-Aldrich C5976) or saline. After preincubation, explants were transferred to fresh incubation media with 10 µm of CL-316,243 and further incubated for 60 min. We measured glycerol in the incubation media using free glycerol reagent (Cat # F6428, Sigma-Aldrich) and Glycerol Standard solution (Cat # G7793, Sigma-Aldrich). To avoid reesterification of fatty acids and glycerol, we included 5 µM of Triacsin C (Cayman Chemical Cat #10007448) in the incubation media. After media collection, we extracted lipids from the fat explants with 1 ml chloroform/methanol (2:1, v/v) and 1% glacial acetic acid for 60 min at 37 °C, and subsequently fat explants were transferred to 500 µl of lysis solution (0.3 N NaOH and 0.1% SDS) and incubated overnight at 55 °C. We measured the protein concentration of these lysates using Pierce 660 nM protein assay reagent (Thermo Scientific) with BSA as standard. Atglistatine was administrated as described[77] dissolved in olive oil (200 µl) at 200 µmol/kg daily for 7 days. Olive oil was used as Vehicle control. After the last dose, BAT was dissected, and we measured ex vivo lipolysis as described above.

For each experiment, we used three biological replicates per group and three technical replicates per biological replicate.

**Plasma lipid clearance and tissue fatty acid uptake and oxidation**. We prepared an emulsion with 20 µl 5% intralipid and 2 µCi (4 µl) [3]H-triolein (Perkin Elmer, Cat # NET43100) per mouse. Volumes were adjusted for the number of mice being studied. Prior to emulsion preparation, [3]H-triolein was evaporated using a nitrogen evaporator. After [3]H-triolein evaporation, we added 5% intralipid and sonicated on ice at low power for 20 s three times. The 5% intralipid/[3]H-triolein emulsion was diluted 1:10 in PBS and 200 µl were injected per mouse to obtain the final concentrations (1 mg of triglycerides and 2 µCi per mouse). We injected the mice with the diluted lipid emulsion via tail vein, collected 10 µl of blood at the time points after injection indicated in Fig. 4g, and added the blood samples to glass scintillation vials containing 5 ml of scintillation cocktail. After the final blood collection, mice were sacrificed. Tissues of interest were dissected and weighed, and a small piece of each tissue was cut, weighed, and put in 750 µl of chloroform/methanol 2:1 for homogenization. After homogenization, 500 µl of 1 M CaCl$_2$ was added, and the samples were centrifuged at $8000 \times g$ for 10 min at 4 °C. The top aqueous layer (containing oxidation products) was transferred to a glass scintillation vial containing 5 ml of scintillation cocktail for measurement. The bottom chloroform phase containing the incorporated lipids was transferred to an empty scintillation vial and evaporated overnight. After evaporation, we added 5 ml of scintillation

cocktail, mixed well, and measured radioactivity with a scintillation counter (Beckman Coulter, LS6000).

**Tissue glucose uptake**. For glucose uptake we performed a modified protocol from ref. [24].

We administered by oral gavage 1 mg/g body weight glucose with 20 µCi deoxy-D-glucose, 2-[1-[14]C]-per mouse as tracer (Perkin Elmer, Cat # NEC495A00). After 1-h, mice were sacrificed and tissues of interest were harvested, weighed, and a small piece was cut, weighed, and solubilized using Solvable$^{TM}$ (0.1 ml per 10 mg of tissue) and 200 µl were added to 5 ml of scintillation cocktail in a glass scintillation vial. Radioactivity was measured using a scintillation counter (Beckman Coulter, LS6000).

**Triglyceride clearance**. For triglyceride clearance, we administered 100 µl of olive oil via oral gavage. We obtained blood samples from the tail vein at the indicated time points in the Fig. 4g, prepared serum, and measured triglycerides, as described above.

**Tissue triglyceride content**. For liver triglyceride content, mice were housed at 23 °C fasted overnight. For BAT triglycerides, mice were housed at 23 °C or exposed to 6 °C overnight and samples were obtained in fed mice. Following euthanization, we weighed total liver and intrascapular BAT and then cut, weighed and snap-froze tissue pieces in liquid nitrogen. Samples were stored at −80 °C and later processed for quantification of tissue triglycerides by the UTSW Metabolic Phenotyping Core as follows: frozen tissue sections (liver ~100 mg and BAT ~50 mg) were homogenized in 1 ml of Folch's solution, using a TissueLyser II bead multi-sample tissue homogenizer (Qiagen Germantown, MD). The homogenate was transferred to a borosilicate glass tube. The original homogenization tube was further rinsed with 1 ml of Folch's solution. The organic extracts were combined, in the borosilicate glass tube, mixed by vortexing and centrifuged in a benchtop centrifuge. The organic extract was transferred to a 5 ml graduated glass flask. The remaining pellet was reextracted and organic extracts were combined. Additional Folch's solution was added to the graduated flask until a volume of 5 ml was reached. Triglyceride levels were determined using a commercial colorimetric enzymatic kit and normalized by tissue weight (Infinity$^{TM}$, Thermo Fisher, Waltham, MA).

**Tissue protein lysate preparation and western blot**. We homogenized tissue (~50 mg per sample) in RIPA buffer containing 50 mM tris(hydroxymethyl)aminomethane, 140 mM sodium chloride, 0.1% sodium dodecyl sulfate, 1% triton X-100, 0.1% sodium deoxycholate, and 0.5 mM ethylene glycol-bis(2-aminoethyl-lether)-N,N,N′,N′-tetraacetic acid, using a TissueLyser II with stainless-steel beads (Qiagen, Germany). After homogenization, we centrifuged the samples at $14,000 \times g$ for 10 min at 4 °C to remove cell debris and collected the supernatants. We measured protein concentration using Pierce® 660 nm Protein assay reagent (Thermo Scientific, Cat # 22660). We mixed the samples with 2× protein sample loading buffer (62.5 mM Tris-HCL, 25% glycerol, 2% SDS, and 0.1% Orange G). For protein electrophoresis, we loaded 20 µg of protein per sample into premade gels [Criterion$^{TM}$ TGX $^{TM}$ 4–20% (Bio-Rad, Cat # 5671094) or AnyKD (Bio-Rad, Cat # 5671124)] and for protein transfer, we used the Criterion Blotter System $^{TM}$ with nitrocellulose membrane (Bio-Rad, Cat # 1620112). We blocked the membranes post-transfer with 5% nonfat dry milk diluted in Tris-buffered saline with 0.1% Tween-20 (TBS-T) for 1 h, and then incubated with the indicated primary antibody with 3% BSA diluted in TBS-T (BSATBS-T) for 12–16 h at 6 °C. After primary antibody incubation, we washed the membrane three times for 5 min each with TBS-T, and then incubated with the appropriate secondary antibody from Li-Cor in BSATBS-T for 30 min. After secondary antibody, we washed membrane three times for 5 min each in TBS-T. We visualized the immunoblotted proteins with the Odyssey CLx near-infrared imaging system (Li-Cor). We provide all the uncropped WB in Supplemetary Figs. 14–19. Antibodies used in this manuscript are shown in Table 1.

**RNA extraction**. We homogenized ~50 mg of tissue in 1 ml of QIAzol (Qiagen, Cat #79306) using a TissueLyser II and stainless-steel beads (Qiagen, 69989). After homogenization, we centrifuged the samples at $14,000 \times g$ for 5 min and then removed floating fat layer from the top by pipetting; we then added 200 ul chloroform and centrifuged the samples at 14,000 g for 15 min. We collected the supernatant and used an RNA extraction kit (Cat #74104, Qiagen) to obtain RNA. During RNA purification we used the RNase-Free DNase Set (Cat #79254, Qiagen) for DNA digestion.

**qPCR**. We prepared cDNA with iScript kit, (Bio-Rad Cat # 1708891) using 1 µg of RNA and followed the manufacturer's instructions, using the following cycles and temperatures: 5 min at 25 °C, 30 min at 42 °C, 5 min at 85 °C and hold at 4 °C.

After the cDNA preparation, we performed qPCR using Power Sybr green (0.1 µM final concentration for primers) on Applied Biosystem's Viaa7 machine.

For qPCR, values are expressed as the mean ± s.e.m. $n = 3–5$ mice/group. Data are representative of at least two independent experiments. Statistical analysis was

**Table 1 Antibodies used for WB.**

| Antibody | Source | Identifier | Figure used | Dilution/use | Raised in |
|---|---|---|---|---|---|
| *Primary antibodies* | | | | | |
| Perilipin 5 | Progen | GP31 | 1e, h, 7b and Supplemetary 1a, 2a, 14a | 1:1000/WB | Guinea pig |
| Perilipin 5 (C-terminal) | Bickel lab[25] | NA | 1c | 1 µg/ml/WB | Rabbit |
| Ucp1 | Abcam | 209483 | 1c and Supplemetary 14a | 1:1000/WB | Rabbit |
| Actin | Santa Cruz Biotechnology | sc-47778 | 1c | 1:1000/WB | Mouse |
| GDI (C-terminal) | Bickel Lab[78] | NA | 1e and Supplementary 1a, 2a, 14a | 1 µg/ml/WB | Rabbit |
| GAPDH | Cell signaling | 2118 S | 1h | 1:1000/WB | Rabbit |
| AKT (total) | Santa Cruz Biotechnology | sc-1619 | 5d, h | 1:1000/WB | Goat |
| pAKT | Cell signaling | 4060 S | 5d, h | 1:1000/WB | Rabbit |
| *Secondary antibodies* | | | | | |
| Donkey anti-guinea pig 800CW | Li-Cor | 926-32411 | 1e, h, 7b and Supplementary 1a, 2a, 14a | 1:15,000 | |
| Goat anti-rabbit 680RD | Li-Cor | 926-68071 | 1c, e, h, 5d, h and Supplementary 1a, 2a, 14a | | |
| Donkey anti-mouse 800CW | Li-Cor | 925-32212 | 1c | | |
| Donkey anti-goat 800CW | Li-Cor | 926-32214 | 5d, h | | |

**Table 2 Primer sequences.**

| Gene | Forward | Reverse | Use |
|---|---|---|---|
| *Plin5* | GAGGCAGCAACAGGGCTACT | CAAAGAGTGTTCATAGGCGAGATG | qPCR |
| *18S* | GAG CGA AAG CAT TTG CCA AG | GGC ATC GTT TAT GGT CGG AA | qPCR |
| *Ucp1* | CCC TGG CAA AAA CAG AAG GA | AGC TGA TTT GCC TCT GAA TGC | qPCR |
| *Elov3* | GCC AAA CTG AAG CAT CCT AAT CTT | CCC AGA ACC ATC TGC AGA ATC | qPCR |
| *Ppargc1a* | TGC CAT TGT TAA GAC CGA G | TTG GGG TCA TTT GGT GAC | qPCR |
| *Dio* | AAG AAG CAC CGG AAC CAA GA | GGC GGC AAG GAG AAA CG | qPCR |
| *Cidea* | GGA GAC CGC CAG GGA CTA C | TTA GTC TGC AAT CCC ATG AAT GTC | qPCR |
| *Prdm16* | TCC GCG TCG AGC AAT AGC | TCT TCC AAG CGG CCA TCA | qPCR |
| *Tfam* | CCAAAAAGACCTCGTTCAGC | ATGTCTCCGGATCGTTTCAC | qPCR |
| *Slc2a4* (Glut4) | GCT TTG TGG CCT TCT TTG AGA | CTG AAG AGC TCT GCC ACA ATG A | qPCR |
| *Fgf21* | GCT GCT GGA GGA CGG TTA CA | CAC AGG TCC CCA GGA TGT TG | qPCR |
| *Cd36* | GGA GGC ATT CTC ATG CCA GT | CTG CTG TTC TTT GCC ACG TC | qPCR |
| *Lpl* | CCA GCT GGG CCT AAC TTT GA | AAC TCA GGC AGA GCC CTT TC | qPCR |
| *Angptl4* | GTT TGC AGA CTC AGC TCA AGG | CCA AGA GGT CTA TCT GGC TCT G | qPCR |
| *Slc27a1* (Fatp1) | TCC TAA GGC TGC CAT TGT GG | GCA CGC ATG CTG TAG GAA TG | qPCR |
| *Il6* | CCA GAG ATA CAA AGA AAT GAT GG | ACT CCA GAA GAC CAG AGG AAA T | qPCR |
| *Adgre1* (F4/80) | CTT TGG CTA TGG GCT TCC AGT C | GCA GGA GGA CAG AGT TTA TCG TG | qPCR |
| *Saa3* | CCT GGG CTG CTG CTA AAG TCA TC | ACC AGG TAG TTG CCC CTC TT | qPCR |
| *Tnfa* | GAG AAA GTC AAC CTC CTC TCT G | GAA GAC TCC TCC CAG GTA TAT G | qPCR |
| *Ccl2* (Mcp1) | AGC ACC AGC CAA CTC TCA C | TCT GGA CCC ATT CCT TCT TG | qPCR |
| *Rsp18* | TGT GTT AGG GGA CTG GTG GAC A | CAT CAC CCA CTT ACC CCC AAA A | qPCR (mtDNA)[74] |
| *Cox2* | ATA ACC GAG TCG TTC TGC CAA T | TTT CAG AGC ATT GGC CAT AGA A | qPCR (mtDNA)[74] |
| *Acss2* | GCT TCT TTC CCA TTC TTC GGT | CCC GGA CTC ATT CAG GAT TG | qPCR[41] |
| *Acaca* | GGA GAT GTA CGC TGA CCG AGA A | ACC CGA CGC ATG GTT TTC A | qPCR[41] |
| *Fasn* | GCT GCG GAA ACT CAG GAA AT | AGA GAC GTG TCA CTC CTG GAC TT | qPCR[41] |
| *Chrebpα* | CGA CAC TCA CCC ACC TCT TC | TTG TTC AGC CGG ATC TTG TC | qPCR[41] |
| *Chrebpβ* | TCT GCA GAT CGC GTG GAG | CTT GTC CGG CAT AGC AAC | qPCR[41] |
| *Dgat2* | AGT GGC AAT GCT ATC ATC ATC GT | TCT TCT GGA CCC ATC GGC CCC AGG A | qPCR |
| *Th* | GAA GGG CCT CTA TGC TAC CCA | TGG GCG CTG GAT ACG AGA | qPCR |
| *Dbh* | GAC TCA ACT ACT GCC GGC ACG T | CTG GGT GCA CTT GTC TGT GCA GT | qPCR |

performed using Student's *t* test, if two groups were analyzed or ANOVA followed by Tukey posttest if more than two groups were analyzed. Comparative Ct method ($\Delta\Delta$ Ct) was used to analyze all qPCR data. Expression was normalized to that of the 18S ribosomal subunit as endogenous control, and the relative expression was calculated in comparison with reference sample that is indicated in each figure.

**Primers.** Primers were designed using Primer Express 3.0.1 (Applied Biosystems) or Primer Blast National Center for Biotechnology Information[79]. All primer sequences are in Table 2. Some primer sequences as indicated in the table were used in previously published studies and referenced.

**General reagents and chemicals.** Tris-glycine-SDS 10× solution, Fisher Bioreagents Cat # BP-1341-4; Tris-glycine 10× solution, Fisher Bioreagents Cat # BP-1306-4; Tris-buffered saline, G Biosciences, Cat # R030; Tween-20, Acros Organics Cat # 23336-0010; methanol, Fisher Chemical Cat # A412SK; BSA, Fisher Scientific Cat # BP16001; free fatty acid BSA, Fitzgerald Cat # 30-AB79; HyClone™ Dulbecco's Modified Eagles Medium, GE Healthcare Life Sciences Cat # SH30022.01; QIAzol Lysis Reagent, Qiagen Cat # 79306; Criterion ™ TGX ™ 4–20%, Bio-Rad Cat # 5671094; Criterion ™ TGX ™ Any Kd, Bio-Rad Cat # 5671124; nitrocellulose membrane, Bio-Rad Cat # 1620112; Power Sybr Green, Applied Biosystems Cat # 4367659; Precision Plus Protein TM Dual Xtra Prestained Protein Standards, Bio-Rad Cat # 1610377; Dulbecco's phosphate-buffered saline, Sigma-Aldrich Cat # D8662; Infinity™ triglycerides liquid stable reagent, Thermo Scientific Cat # TR22421; Multi calibrator lipids, Wako Cat # 464-01601; HR series NEFA-HR, Wako Cat # 999-34691; NEFA standard solution, Wako Cat # 276-76491; free glycerol reagent, Sigma-Aldrich Cat # F6428; Glycerol Standard solution, Sigma-Aldrich Cat # G7793; Pierce 660 nM Protein Assay Reagent, Thermo Scientific Cat # 22660; 3a70B Complete Counting cocktail, RPI Cat # 111154; Solvable ™, Perkin Elmer Cat # 6NE9100; paraformaldehyde, Sigma-Aldrich Cat #

P6148; sodium pyruvate, Sigma-Aldrich Cat # P5280; L-(−)-malic acid, Sigma-Aldrich Cat # M1000; GDP sodium salt, Sigma-Aldrich Cat # G7127; D-(+)-glucose, Sigma-Aldrich Cat # G7528; D-sucrose, Fisher Bioreagents Cat # BP220-1; D-mannitol, Sigma-Aldrich Cat # M4125; HEPES 1 M solution, Sigma-Aldrich Cat # H0887; EGTA, MP Biomedicals Cat # 195174; potassium phosphate monobasic ($KH_2PO_4$), Sigma-Aldrich Cat # P5655; magnesium chloride ($MgCl_2$), Sigma-Aldrich Cat # M8266; Tris base, Fisher Bioreagents Cat # BP152-1; sodium chloride ($NaCl_2$), Fisher Scientific Cat # S671-3; sodium deoxycholate, Sigma-Aldrich Cat # 30970; Tris-HCl 1 M solution, Fisher Bioreagents Cat # BP1757; glycerol, Acros Organics Cat # 327255000; orange G, Sigma-Aldrich Cat # O3756; sodium hydroxide (NaOH), Fisher Scientific Cat # S318-1; sodium dodecyl sulfate (SDS), Invitrogen Cat # 15-525-017; chloroform, Sigma-Aldrich, Cat # C24322; mercaptoethanol, Fisher Scientific Cat # 03446I-100; glacial acetic acid, Fisher Bioreagents Cat # BP2401.

**Statistics and reproducibility**. For statistical analysis, we used GraphPad Prism version 7.00 for MacOS, GraphPad Software, La Jolla California USA, www.graphpad.com.

For all the experiments, data are representative of at least two independent experiments and all attempts to reproduce were successful. $P$ values are indicated in figures or figure legends, and statistical analysis was performed using Student's $t$ test if two groups were analyzed or ANOVA followed by Tukey posttest if more than two groups were analyzed. Statistical significance is defined as $p < 0.05$.

**Software**. For WB band intensity analysis, we used Image Studio Ver. 3.1 (Licor Biosciences). For qPCR Ct values analysis, we used Quant Studio Real-time qPCR software Ver. 3.1 (Applied Biosystems). Mitochondrial respiration data were analyzed using the Wave Desktop Software Ver. 2.6.1(Agilent technologies). For adipocyte area calculation, we used Keyence BZ-X Analyzer software Ver. 1.3.03. For colorimetric microplate assays (protein quantification, glycerol, and citrate synthase activity), we used Gen5 Ver 2.01.14 software. For quantification of LD–mitochondria contact sites, mitochondria length, and aspect ratio, we used Image J software NIH Ver. 1.53.

**Reporting summary**. Further information on research design is available in the Nature Research Reporting Summary linked to this article.

## Data availability

The authors declare that all the data supporting the findings of this study are available within the article and its Supplementary Data, or from the corresponding author upon request. Source data are provided with this paper.

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

## Acknowledgements

We dedicate this manuscript to the late Lisa Hahner, who contributed so much her great skill and effort to this project. We thank Philipp Scherer and UT Southwestern Touchstone Diabetes Center for the UCP1rtTA mouse, and his help designing the TRE-Plin 5 transgene. We thank the UT Southwestern Metabolic Phenotyping Core (Ruth Gordillo and Syann Lee), Electron Microscopy Core (Kate Luby-Phelps and Anza Dar-ehshouri), Histo Pathology Core (Bret M. Evers and John M. Shelton), and Transgenic Core (Robert E. Hammer). This study was supported by NIH R01DK115875 for P.E.B., NIH R01DK112826 and R01DK108833 for W.L.H., and Wayne State University Startup funds for R.F.-V.

## Author contributions

V.I.G.M. conceived the study, carried out experiments, interpreted the data, and wrote the manuscript; C.Y. designed and generated mouse models, and contributed to writing the methods; L.H. assisted with the execution of experiments, genotyped and maintained mice colony, and contributed to writing the methods; J.L.M. assisted with the execution of experiments; R.F.-V. designed the construct for Perilipin 5 KO mouse line; J.A.J. carried out fatty acid uptake and oxidation experiments; W.L.H. assisted with design and interpretation of fatty acid uptake and oxidation experiments; P.E.B conceived and supervised the study, interpreted the data, wrote, and revised the manuscript. All the authors discussed the results and commented on the manuscript.

## Competing interests

The authors declare no competing interests.
