## [Peer Review File · Nature Communications]

Reviewer comments, first round –

Reviewer #2 (Remarks to the Author):

Montejano et al. have reported exciting new findings on the role of Perilipin 5 in brown adipose tissue which drive improvements in systemic energy homeostasis. Building on previous work uncovering a transcriptional regulatory axis for PLIN5 through SIRT1 and PGC-1 α in cells, the authors now utilize several tissue specific mouse models, including crosses onto the Ucp1 KO background, to understand the PLIN5 impact on in vivo thermogenic function. Overall, the work is comprehensive and furthers our understanding of thermogenic machinery. In order to be even more appropriate for Nature Communications, the following comments should be considered.

Major comments

1. The cold induction of PLIN5 at the transcriptional and protein levels is very clear. Is this recapitulated by norepinephrine or CL-316,243 in cells or animals?
2. PLIN5 has been shown in muscle and cardiomyocytes to be associated with the mitochondria and provide a contact site between mitochondria and lipid droplets. Does a similar function occur in brown adipocytes?
3. How do the current findings integrate into the previous work from the authors on PLIN5 transcriptional regulation? To that end, global RNA profiling would be a major improvement over the few select genes in figure 3e and would give an important overview of the total PLIN5 dependent transcriptome. On a related note, despite the histology revealing no differences in BATiPLIN5 mice versus controls at 30. degrees C, does PLIN5 overexpression still increase gene expression or other parameters at thermoneutrality? Thermoneutrality offers the possibility to parse out the purely transcriptional elements of PLIN5 control versus its more canonical role in lipid storage/mobilization.
4. On a related note, can the increased mitochondrial cristae density be recapitulated in vitro? If so, could the author's target lipolysis or SIRT1 to interrogate whether the mito effects require the lipid droplet-associated function or the transcriptional action of PLIN5, respectively.
5. The results in the iWAT are intriguing from a standpoint that the white adipocytes from BATiPLIN5 mice appear healthier and more insulin sensitive yet have reduced beiging. Are there changes in sympathetic tone (e.g. TH staining or qPCR) in the iWAT depot?

Minor comments

1. What are the serum lipid levels during the cold study in Figure 3?
2. How do the authors propose the improved glucose clearance is being mediated if more glucose is not being taken up by the BAT or any other surveyed tissues in the BATiPLIN5 mice (Supp Fig 2b)?
3. What is the author's rationale for why 3c shows constitutive difference in body temp defense and in 3d the difference only occurs after a few hours? Isn't the only distinction the removal of food at day 21?

Reviewer #3 (Remarks to the Author):

Perilipin 5 (PLIN5), a member of the Perilipin family of lipid droplet-associated proteins, is expressed in highly oxidative tissues such as heart, skeletal muscle and brown adipose tissues. Previous studies have shown that Plin5 plays a role in regulating mitochondria activity in muscle cells and in adipocytes. Plin5 deficient mice also showed insulin resistance. To elucidate the functions of Plin5 in BAT, authors generated three animal models including an inducible, BAT-specific Plin5 overexpression in wild type mice (BATiPLIN5), in UCP1 KO mice (BATiPLIN5/UCP1 KO) and animals with BAT-specific knockout of Plin5 (BKOPLIN5). Authors observed that overexpression of PLIN5 in BAT resulted in an increased body-temperature, denser mitochondrial cristae structure and enhanced respiratory function upon cold exposure. Overexpression of PLIN5 in BAT also led to improved insulin sensitivity. However, animals with BAT-specific KO of Plin5 did not show obvious phenotype in mitochondrial function and body temperature regulation. These data are helpful to understand the role of Plin5 in controlling BAT functions.

Main concerns:

- 1) As the role of Plin5 in controlling mitochondrial activity and insulin sensitivity was previously shown in other tissues, the novelty of current manuscript remains unclear. No further mechanistic insight regarding the role of Plin5 was provided in current manuscript.
- 2) Majority of data (Fig. 1-6) in the manuscript were from animals of overexpressing Plin5 in BAT (BATiPLIN5). Some of the phenotypes were not correlated with that obtained from animals with BAT-specific KO of Plin5.
- 3) In the manuscript, overexpression or KO of Plin5 in BAT was driven by Ucp-1 promoter. As Ucp1 was also shown to be induced in subcutaneous WAT (SWAT) upon cold induction, it is important to check the phenotype of those animals in SWAT.

Minor comments

- 1) In figure 1, Plin5 expression is induced in BAT by cold stimuli. How about its expression levels in SWAT and gWAT, heart and skeletal muscle? These data are important in understanding the precise role of Plin5.
- 2) The quality of mitochondrial images by EM need to be improved and quantitative analyses may help to evaluate the structural difference.
- 3) In figure 4, the rate of lipolysis was not changed in BATiPLIN5 mice under basal or stimulated condition. The lipid uptake and oxidation was increased in the BAT at 23 °C. How about under cold exposure?
- 4) Page 16 Line 17, "Supplementary Figure 5b and c" should be "Supplementary Figure 5a and b"? There is no Supplementary Figure 5c.
- 5) Large amount of data was included in each Figure and the logic appears to be confusing.

AUTHORS' RESPONSES TO REVIEWER COMMENTS

Reviewer #2 (Remarks to the Author):

Montejano et al. have reported exciting new findings on the role of Perilipin 5 in brown adipose tissue which drive improvements in systemic energy homeostasis. Building on previous work uncovering a transcriptional regulatory axis for PLIN5 through SIRT1 and PGC-1alpha in cells, the authors now utilize several tissue specific mouse models, including crosses onto the Ucp1 KO background, to understand the PLIN5 impact on in vivo thermogenic function. Overall, the work is comprehensive and furthers our understanding of thermogenic machinery. In order to be even more appropriate for Nature Communications, the following comments should be considered.

RESPONSE: We greatly appreciate Reviewer #2's comments that the findings we are reporting are exciting, new, and comprehensive and advance the field of adaptive thermogenesis. We also appreciate Reviewer #2's insightful critiques and suggestions for improvement of our manuscript, which have led us to perform new experiments that we report below.

Major comments

1. The cold induction of PLIN5 at the transcriptional and protein levels is very clear. Is this recapitulated by norepinephrine or CL-316,243 in cells or animals?

RESPONSE: We thank the reviewer for posing this question, which is important given the critical role of the sympathetic nervous system in activation of thermogenic adipose tissue function. To answer this question, we injected 12-week-old male mice (C56BL/6J) intraperitoneally with either Vehicle or the β -3 adrenergic receptor agonist CL-316,243 daily for 2 days or 7 days. At those time points, we harvested the BAT and assayed for PLIN5 protein by immunoblot (Supplementary Figure 1a) and *Plin5* mRNA by qPCR (Supplementary Figure 1b). We observed higher expression of both PLIN5 protein and mRNA at both time points. The highest level of PLIN5 protein was at 2 days, which is similar to the induction pattern of PLIN5 protein in BAT during cold exposure (Figure 1h). We show these new data in Supplementary Figure 1a,b and discuss the results in the revised main text.

2. PLIN5 has been shown in muscle and cardiomyocytes to be associated with the mitochondria and provide a contact site between mitochondria and lipid droplets. Does a similar function occur in brown adipocytes?

RESPONSE: Thank you for raising this important point. We have taken additional EM pictures and quantified LD/mitochondria contact sites using the methods described by Benador et al. (Main text Reference # 56) in our models of BAT-specific PLIN5 overexpression (BATiPLIN5) and deficiency (BKOPLIN5) housed at 23 °C and 6 °C. In agreement with the findings of Benador et.al., we observed a decrease in LD/mitochondria contact sites in the BAT of Control mice during cold exposure

compared to room temperature; whereas, with PLIN5 overexpression, there no significant change in the % mitochondria in contact with LD between housing temperatures (Supplementary Figure 7a). There was a statistically significant increase in LD/mitochondria contacts in BATiPLIN5 mice compared with in Controls at 6 °C. However, the extent of those contacts as assessed by contact surface normalized to either mitochondrial perimeter or LD perimeter was unchanged, perhaps suggesting that these contacts were mostly “kissing” rather than “hugging.” However, in mice maintained at room temperature we observed a statistically significant decrease in contact sites between lipid droplets and mitochondria in the BATiPLIN5 versus Control mice. This last result is contrary to other models in which it has been reported that increased PLIN5 is associated with an increase in LD/mitochondria contact sites (Bosma et al and Wang et al. Main text References # 30 and 31). This may reflect a fundamental difference in PLIN5 function in different tissues.

With respect to our model of PLIN5 deficiency in BAT (BKOPLIN5), we did not observe differences in LD/mitochondria contact sites between our PLIN5 KO model at 23 °C and 6 °C (Supplementary Figure 12). Our data suggest that Perilipin 5 is not required for LD/mitochondria contact in BAT. However, due to the decrease in mitochondria/LD contact sites in the BATiPLIN5 mice at room temperature compared with Controls, our data are consistent with a role for PLIN5 in shifting BAT mitochondria to a more oxidative state, inasmuch as Benador et al. reported that mitochondria that are not in contact with lipid droplets are more oxidative versus glycolytic. We discuss these data in the main text. The authors are very appreciative of this question raised by the reviewer, as the additional experiments suggested have increased the rigor, quality and significance of our manuscript.

3. How do the current findings integrate into the previous work from the authors on PLIN5 transcriptional regulation? To that end, global RNA profiling would be a major improvement over the few select genes in figure 3e and would give an important overview of the total PLIN5 dependent transcriptome.

RESPONSE: We agree that global RNA profiling will be an important method to generate hypotheses regarding the actions of PLIN5 in brown adipocytes. For this manuscript we decided to focus on the effects of PLIN5 on gene expression related to thermogenesis and to specific aspects of lipid and glucose metabolism, which are major aspects the phenotypes we have observed. Regarding integration with our prior work, we have examined expression of PGC1 α target genes (Figures 3e and 7j), and we have examined the effects of SIRT1 inhibition on the mitochondrial phenotypes of BATiPLIN5 mice (Figure 3e,f and Supplementary Figures 9 and 10). We are generating mouse models that build on previous work specifically in regard to PGC1 α (e.g. effect of inducible knockout of PGC1 α in BAT on the BATiPLIN5 phenotypes), and we look forward to reporting the resulting data in future manuscripts.

Reviewer Comment: On a related note, despite the histology revealing no differences in BATiPLIN5 mice versus controls at 30 °C, does PLIN5 overexpression still increase gene expression or other parameters at thermoneutrality? Thermoneutrality offers the

possibility to parse out the purely transcriptional elements of PLIN5 control versus its more canonical role in lipid storage/mobilization.

RESPONSE: We agree with Reviewer 2 that experiments at thermoneutrality would be very informative if the *Plin5* transgene were expressed in BAT at 30°C. Unfortunately, with the BATiPLIN5 model, which is dependent in the UCP1 promoter, we do not observe overexpression when housed at thermoneutrality. To make this point clear, we now show a western blot that demonstrates no significant increase in expression of PLIN5 at thermoneutrality in BATiPLIN5 mice compared with Control mice (Supplementary Figure 1c). To address the important issue of transcriptional effects versus canonical effects of PLIN5, we will pursue experiments in a mouse model that overexpresses Perilipin 5 in BAT in a temperature-independent manner (Adiponectin promoter).

4. On a related note, can the increased mitochondrial cristae density be recapitulated *in vitro*? If so, could the author's target lipolysis or SIRT1 to interrogate whether the mito effects require the lipid droplet-associated function or the transcriptional action of PLIN5, respectively.

RESPONSE: We thank the reviewer for suggesting this important experiment. As we have been working extensively with our *in vivo* mouse models and as the mitochondrial cristae phenotype is consistent in the BATiPLIN5 mice, we have used our *in vivo* model to test this hypothesis. We treated BATiPLIN5 and Control mice housed at 23 °C with intraperitoneal injections of Atglistatine (ATGL inhibitor) or EX-527 (SIRT1 inhibitor) or the respective Vehicles. First, we acknowledge that inhibition of ATGL with Atglistatine is systemic and does not specifically target BAT lipolysis, nor does it specifically target lipolysis in BAT that is coordinated by PLIN5 on account of the co-expression of PLIN1 in this tissue. With these caveats, we first confirmed with assays of lipolysis and of SIRT1 deacetylase activity that the systemic administration of Atglistatine or EX-527 decreased ATGL or SIRT1 function, respectively, in BAT (Supplementary Figure 9a and 9b). Then we harvested BAT from the treated mice and prepared sections for analysis by transmission electron microscopy. As before, in the mice receiving Vehicle, we observed an increase in total cristae length normalized to mitochondrial area in BATiPLIN5 mice compared with that of Control mice (Figure 6f). This increase was attenuated significantly in mice receiving IP injections of either Atglistatine or EX-527. We now include electron micrographs, as well quantification of cristae length in Figures 6e and f and Supplementary Figure 10, and we also discuss these results in the text. These results suggest that the effects of PLIN5 expression on mitochondria require its actions in the nucleus as a transcriptional regulator and intact lipolytic machinery, which is consistent with a recent report from Doug Mashek's lab (Najt et al. Main text Reference # 64). The Mashek lab has elegantly demonstrated in hepatocytes that PLIN5 binds and carries monounsaturated fatty acids derived from LD lipolysis to the nucleus to deliver the FA to SIRT1 to allosterically promote its activity. We are pursuing studies with additional genetically modified mice that target ATGL, SIRT1 or PGC-1 α in BAT in order to extend our pharmacological data with genetic approaches.

5. The results in the iWAT are intriguing from a standpoint that the white adipocytes from BATiPLIN5 mice appear healthier and more insulin sensitive yet have reduced beigeing. Are there changes in sympathetic tone (e.g. TH staining or qPCR) in the iWAT

RESPONSE: We agree that these findings are intriguing and thank Reviewer 2 for their suggested experiment. We measured mRNA levels of tyrosine hydroxylase (TH) and dopamine- β -hydroxylase (DBH), which are required for the biosynthesis of catecholamines. We observed an increase in expression of these 2 enzymes with cold exposure in both Control and BATiPLIN5 mice, but we did not find differences in their mRNA expression between the two genotypes at either 23 or 6 °C. Now we include this result in Supplementary Figure 4b and discuss it in the text. These data are consistent with our finding that beigeing genes are not induced in the iWAT of BATiPLIN5 mice. Other possible mechanisms for the healthy remodeling of iWAT and improvement in glucose tolerance of the BATiPLIN5 mice may involve differential BATokine or lipokine secretion. We have ruled out FGF21 as a likely mediator on the basis of circulating levels (Supplementary Figure 4b); identification of potential mechanisms remains for future studies.

Minor comments

1. What are the serum lipid levels during the cold study in Figure 3?

RESPONSE: For this particular experiment we did not measure the serum lipid levels. In Figure 4 we presented lipid levels after housing at different temperatures. These measurements were determined in fed mice and not during fasting as in the cold tolerance experiments of Figure 3. We observed a trend to lower NEFA levels in the BATiPLIN5 mice at 23 °C and almost significant $p=0.06$ at 4 °C.

2. How do the authors propose the improved glucose clearance is being mediated if more glucose is not being taken up by the BAT or any other surveyed tissues in the BATiPLIN5 mice (Supp Fig 2b)?

RESPONSE: Based on the improvement in pyruvate tolerance observed in BATiPLIN5 mice, one possible explanation is that BATiPLIN5 mice have reduced gluconeogenesis in the liver. It is also possible that we also are missing increased glucose uptake that would be more evident in a glucose clamp or with more exhaustive harvesting of muscle tissues.

3. What is the author's rationale for why 3c shows constitutive difference in body temp defense and in 3d the difference only occurs after a few hours? Isn't the only distinction the removal of food at day 21?

RESPONSE: We noticed this discrepancy as well. On days 2, 4, 14, and 21 of Figure 3c, temperature was measured with food in the cage and without disturbing the mice. For time 0 of Figure 3d the temperature was measured immediately after removing the food. We think that certain level of stress in the mice have played a role in the

discrepancy in this experiment. However, in independent experiments we have found data that are similar to that shown in Figure 3c.

Reviewer #3 (Remarks to the Author):

Perilipin 5 (PLIN5), a member of the Perilipin family of lipid droplet-associated proteins, is expressed in highly oxidative tissues such as heart, skeletal muscle and brown adipose tissues. Previous studies have shown that Plin5 plays a role in regulating mitochondria activity in muscle cells and in adipocytes. Plin5 deficient mice also showed insulin resistance. To elucidate the functions of Plin5 in BAT, authors generated three animal models including an inducible, BAT-specific Plin5 overexpression in wild type mice (BATiPLIN5), in UCP1 KO mice (BATiPLIN5/UCP1 KO) and animals with BAT-specific knockout of Plin5 (BKOPLIN5). Authors observed that overexpression of PLIN5 in BAT resulted in an increased body-temperature, denser mitochondrial cristae structure and enhanced respiratory function upon cold exposure. Overexpression of PLIN5 in BAT also led to improved insulin sensitivity. However, animals with BAT-specific KO of Plin5 did not show obvious phenotype in mitochondrial function and body temperature regulation. These data are helpful to understand the role of Plin5 in controlling BAT functions.

RESPONSE: The authors appreciate Reviewer #3's comment that our manuscript will assist the field in understanding how PLIN5 regulates BAT functions. We emphasize that mice with BAT-specific KO of *Plin5* do demonstrate a phenotype in mitochondrial function (new Figure 7e,f). Specifically, OCR is reduced in mitochondria isolated from the BAT of BKOPLIN5 mice at 23 °C (Figure 7f), even without clear morphological changes in mitochondria at that housing temperature (Figure 7e). Also, we now present new quantitative data that, when challenged at 6 °C, BKOPLIN5 mice have reduced total cristae length normalized to mitochondrial area in BAT compared with Control mice, and the morphological differences in cristae between Control and BKOPLIN5 mice visually striking (Figure 7e). With respect to body temperature regulation, the BKOPLIN5 mice are likely able to maintain body temperature in the cold to the same degree as Control mice due to compensatory activation of the beige program in inguinal WAT, as suggested by induction of *Ucp1* and *Pparg1a* gene expression (Figure 7j).

Main concerns:

1) As the role of Plin5 in controlling mitochondrial activity and insulin sensitivity was previously shown in other tissues, the novelty of current manuscript remains unclear. No further mechanistic insight regarding the role of Plin5 was provided in current manuscript.

RESPONSE: We agree that metabolic phenotypes of PLIN5 gain- and loss-of-function in tissues other than BAT have been published by other labs (Main text References # 32,35,36,37,38). Nevertheless, given that BAT is a tissue specialized for adaptive

thermogenesis, we suggest that the function of PLIN5 in BAT likely differs from its function in liver, heart, and skeletal muscle. In fact, the function of PLIN5 likely depends on the metabolic function of each tissue and on the tissue expression of other Perilipins, especially of Perilipin 1 (PLIN1). For example, in the heart (high PLIN5, no PLIN1), PLIN5 overexpression results in steatosis due to inhibition of lipolysis (Pollak et al. Main text Reference # 36); whereas, in BAT (high PLIN5, high PLIN1), we find that overexpression of PLIN5 does not result in steatotic BAT at 23 °C (Figure 4a,b) and does not significantly alter basal or stimulated lipolysis (Figure 4c). Also, overexpression of PLIN5 in skeletal muscle has been associated with increased glucose tolerance as attributed to markedly elevated circulating FGF21 (Harris et al. Main text Reference # 50), but mice with PLIN5 overexpression in BAT have improved systemic glucose tolerance with unchanged circulating FGF21 levels (Supplementary Figure 4c). The mechanism of action must differ between BAT and skeletal muscle. Further, no prior work by any lab has demonstrated the healthy remodeling of inguinal white adipose tissue in response to altered PLIN5 expression in any different tissue. Our cumulative data suggest the exciting hypothesis that BAT has the potential to secrete a protein, lipid or other metabolite that promotes healthy white fat remodeling with smaller adipocytes, lower expression of some inflammatory markers, increased *Glut4* expression, and improved insulin sensitivity (Figure 5). We remain confident that our manuscript reports a series of novel findings that bear on the function of PLIN5 specifically in BAT and on its role in systemic glucose and lipid metabolism that merit publication in *Nature Communications* and that will move the field of thermogenic adipose tissue forward.

1. This is the first manuscript to report the assessment of PLIN5 function in BAT by tissue-specific gain- and loss-of-function experiments in mice.
2. PLIN5 has been implicated in the formation LD/mitochondrial contact sites in other tissues and cells to the point of being referred to as a LD/mitochondrial “tether” (Bosma et al and Wang et al. Main text References # 30 and 31). However, BKOPlin5 mice, which completely lack PLIN5 in brown adipocytes, did not show any differences in LD/mitochondrial contact sites. Also, contrary to previous reports in other tissues or cell culture, PLIN5 overexpression in BAT of mice housed at 23 °C does not increase LD/mitochondrial contact sites but rather reduces them, which is consistent with its role in promoting fatty acid oxidation in BAT, as we discuss in the manuscript.
3. We also report that the major phenotypes in BATiPLIN5 mice with respect to systemic glucose tolerance and healthy WAT remodeling are UCP1-dependent, which suggests at a minimum that these phenotypes depend on fully functional mitochondria in BAT.
4. Two additional mechanistic features for the mitochondrial phenotype of PLIN5 deficiency in BAT are the requirement on intact lipolysis and SIRT1 activity.

2) Majority of data (Fig. 1-6) in the manuscript were from animals of overexpressing Plin5 in BAT (BATiPLIN5). Some of the phenotypes were not correlated with that obtained from animals with BAT-specific KO of Plin5.

RESPONSE: Thank you for raising this issue so that we may make our findings and interpretations more clear. Given the increased glucose tolerance and cold tolerance we observed in BATiPLIN5 mice, one might expect that BKOPLIN5 mice would have worse glucose tolerance and cold tolerance compared with Controls, which they did not. In this regard, we found compensatory increased expression of thermogenic genes in the iWAT of the BKOPLIN5 mice. This compensation likely was due to the defects in mitochondrial form and function in BAT that we observed in BKOPLIN5 mice, thereby leading to defective BAT thermogenic function and compensatory iWAT beiging. Such iWAT beiging would be expected to balance the defects in BAT thermogenesis and glucose metabolism. We are pursuing this hypothesis through a novel inducible BAT-specific PLIN5 mouse model, and we are in the early stages of phenotyping. The idea is to assess cold tolerance in the acute setting of *Plin5* gene knockout prior to iWAT compensation via beiging. Preliminary data show reduced body temperature in the BKOPLIN5 mice relative to Control mice by 8 hours of acute cold challenge. We look forward to reporting this data in a fully developed manuscript.

3) In the manuscript, overexpression or KO of *Plin5* in BAT was driven by *Ucp-1* promoter. As *Ucp1* was also shown to be induced in subcutaneous WAT (SWAT) upon cold induction, it is important to check the phenotype of those animals in SWAT.

RESPONSE: In our BATiPLIN5 model we do not observe overexpression of *Plin5* mRNA (including transgene and wildtype *Plin5* mRNA) in iWAT at 23 °C or after short term cold exposure relative to Control mice (Figure 5b). Nevertheless, we did extensive phenotyping of inguinal WAT, which is shown in Figure 5, including adipocyte size, gene expression, insulin sensitivity, and basal and catecholamine stimulated lipolysis. We think that iWAT phenotypes we report are independent of PLIN5 expression in iWAT but rather represent a secondary effect of PLIN5 overexpression in BAT. Because BAT and WAT have differential functions and because PLIN5 also increases with cold in the iWAT of Control mice, as showed in Figure 5b, we think that additional studies beyond the scope of this manuscript must be performed in models that overexpress or knockout PLIN5 exclusively in white adipose tissue. The problem is that this WAT-specific manipulation of PLIN5 gene expression will require novel mouse models that have not been reported to date.

Minor comments from Reviewer #3

1) In figure 1, *Plin5* expression is induced in BAT by cold stimuli. How about its expression levels in SWAT and gWAT, heart and skeletal muscle?

RESPONSE: Thank you for raising this question. We did not prepare a particular Figure for this question, but relevant information is available in Figure 5b and 7b. Following 16h at 6 °C, *Plin5* gene expression in iWAT increases just under 15-fold (Figure 5b) but remains unchanged in gWAT (Supplementary Figure 4b). At the level of protein expression, PLIN5 levels do not change significantly in heart or soleus muscle after 16h at 6 °C (Figure 7b). We consider it very important to study the function of PLIN5 in iWAT in thermogenesis, and this is a goal of ongoing studies in our lab.

2) The quality of mitochondrial images by EM need to be improved and quantitative analyses may help to evaluate the structural difference.

RESPONSE: We have acquired new EM images on a new electron microscope to address the issue of image quality. We note that some images of mitochondria from BATiPLIN5 mice appear to lack resolution of the cristae, but this is not an issue of image quality but rather cristae density. In the image below, increasing the size of the image permits resolution of individual cristae sufficient to permit morphometric analysis with ImageJ, and these data are reported. In our revised manuscript, we include Crista length measurement as well lipid droplet/mitochondria contact measurements.

3) In figure 4, the rate of lipolysis was not changed in BATiPLIN5 mice under basal or stimulated condition. The lipid uptake and oxidation was increased in the BAT at 23 °C. How about under cold exposure?

RESPONSE: We did not measure lipolysis or lipid uptake and oxidation under conditions of cold exposure, though these will be interesting studies for future investigations. That the rate of lipolysis was unchanged in BAT of BATiPLIN5 mice does not mean that lipolysis in other tissues, such as WAT or the heart did not change. In fact, catecholamine-stimulated lipolysis was increased in the iWAT of BATiPLIN5 mice, as shown in Figure 5e.

4) Page 16 Line 17, “Supplementary Figure 5b and c” should be “Supplementary Figure 5a and b”? There is no Supplementary Figure 5c.

RESPONSE: Thank you for pointing out this error. We have corrected the text.

5) Large amount of data was included in each Figure and the logic appears to be confusing.

RESPONSE: Thank you for this feedback. We hope the changes we have made to the Figures have improved the clarity and logic of the manuscript.

Reviewer comments, second round –

Reviewer #2 (Remarks to the Author):

The authors have provided comprehensive explanations and experimental evidence addressing my previous points. I have no further concerns or questions that are within the scope of the current work.

Reviewer #3 (Remarks to the Author):

The authors have addressed most of the concerns. While I appreciate the efforts, I still feel that there are some key points missing in this manuscript. It is well known that PLIN5 is a lipid droplet protein, but it is reported to affect the mitochondrial cristae density. The authors show that PLIN5 does not affect the contact sites between lipid droplet and mitochondria, but works through ATGL and SIRT1 to exert the function. The mechanism of PLIN5 is still confusing in the current version. Also, it is still not clear how PLIN5 in BAT remodels inguinal white adipose tissue. The authors suspect that there might be a secretory factor mediating the effect, but there is no evidence.

REVIEWERS' COMMENTS

Reviewer #2 (Remarks to the Author):

The authors have provided comprehensive explanations and experimental evidence addressing my previous points. I have no further concerns or questions that are within the scope of the current work.

Response:

We thank Reviewer #2 for their time spent reviewing our manuscript and for helping us make it better. We thank them for their positive response to our revised manuscript and to our replies to their prior

Reviewer #3 (Remarks to the Author):

The authors have addressed most of the concerns. While I appreciate the efforts, I still feel that there are some key points missing in this manuscript. It is well known that PLIN5 is a lipid droplet protein, but it is reported to affect the mitochondrial cristae density. The authors show that PLIN5 does not affect the contact sites between lipid droplet and mitochondria, but works through ATGL and SIRT1 to exert the function. The mechanism of PLIN5 is still confusing in the current version. Also, it is still not clear how

PLIN5 in BAT remodels inguinal white adipose tissue. The authors suspect that there might be a secretory factor mediating the effect, but there is no evidence.

Response:

We also thank Reviewer #3 for their time spent reviewing our manuscript and for helping us improve it. We also appreciate their positive assessment that we have addressed most of their concerns in our revised manuscript and in our replies to their prior critiques. We acknowledge that we have not answered all mechanistic questions that our work has raised, and Reviewer #3 has appropriately noted these questions. We view the raising of such new questions to be a major contribution of this work to the field. We have cited our prior work about activation of the SIRT1/PGC1 α gene program by PLIN5 and the corresponding downstream effects on gene expression related to mitochondrial biogenesis and function (Gallardo et al. NComm, 2016), and we have shown in the current manuscript *in vivo* that PLIN5 expression is associated with changes in expression of UCP1 and PGC1 α , mitochondrial cristae packing, and mitochondrial respiratory function. Though we think it highly likely that much of the positive effect of PLIN5 on mitochondrial respiratory function is mediated by SIRT1 via the SIRT1/PGC1 α gene program, we cannot rule out a direct effect of PLIN5 on mitochondria by physical association. We clearly state this caveat in our new last paragraph of the Discussion that enumerates limitations of our current study. There is more ambiguity about the role of ATGL, but the work of Mashek's lab, as we have cited (Ref. 58) has indicated that PLIN5 shuttles a fatty acid to SIRT1 to activate it. As we note in the Discussion, the role of ATGL may be in non-BAT tissues, such as WAT or

heart, or if in BAT, more likely involves PLIN1, as we have not been able to identify measurable differences in lipolysis in BAT explants from BATiPLIN5 and Control mice. It is likely that in BAT PLIN5 does not work through ATGL, but nevertheless depends on intact ATGL lipolytic function to provide the requisite fatty acid for SIRT1 activation.

Also, in our paragraph on limitations, we note that we have not identified the mechanism by which PLIN5 in BAT exerts effects on WAT adipocyte size and function (insulin sensitivity and lipolysis). We we have ruled out two of the usual suspects (increased WAT beiging and increased serum FGF21), but clearly much additional work needs to be done, including unbiased approaches. We have hypothesized that a secreted factor from BAT may be responsible, but this could be a protein, lipid or other metabolite. As we continue to work with our mouse models to identify candidate secreted factors, we request Reviewer #3's forbearance on this issue.